# TOWARDS GRAPH-LEVEL ANOMALY DETECTION VIA DEEP EVOLUTIONARY MAPPING

## ABSTRACT

Graph-level anomaly detection aims at capturing anomalous individual graphs in a graph set. Due to its significance in various real-world application fields, such as identifying rare molecules in chemistry and detecting potential frauds in online social networks, graph-level anomaly detection has received great attention. In distinction from node- and edge-level anomaly detection that is devoted to identifying anomalies on a single graph, graph-level anomaly detection faces more significant challenges because both the *intra-* and *inter-*graph structural and attribute patterns need to be taken into account to distinguish anomalies that exhibit deviating structures, rare attributes or the both. Although deep graph representation learning shows effectiveness in fusing high-level representations and capturing characters of individual graphs, most of the existing works are defective in graph-level anomaly detection because of their limited capability in exploring information across graphs, the imbalanced data distribution of anomalies, and low interpretability of the black-box graph neural networks (GNNs). To overcome these limitations, we propose a novel deep evolutionary graph mapping framework named GmapAD, which can adaptively map each graph into a new feature space based on its similarity to a set of representative nodes chosen from the graph set. By automatically adjusting the candidate nodes using a specially designed evolutionary algorithm, anomalies and normal graphs are mapped to separate areas in the new feature space where a clear boundary between them can be learned. The selected candidate nodes can therefore be regarded as a benchmark for explaining anomalies because anomalies are more dissimilar/similar to the benchmark than normal graphs. Through our extensive experiments on nine real-world datasets, we demonstrate that exploring both *intra-* and *inter-*graph structural and attribute information are critical to spot anomalous graphs, and our framework outperforms the state of the art on all datasets used in the experiments[1].

## 1 INTRODUCTION

Graph-level anomalies are abnormal or rare individual graphs in a graph set. These anomalies can be observed in various application fields, such as rare molecules and abnormal proteins in biochemistry, brain disorders in brain networks/graphs, and frauds in online social networks (Noble & Cook, 2003; Akoglu et al., 2015). Detecting this category of anomalies has shown great benefits in facilitating downstream anomaly handling process, alleviating anomalies' detrimental impact on society, and boosting real-world applications (e.g., health monitoring and drug discovery). However, graph-level anomaly detection differs significantly from node- and edge-level anomaly detection that investigates an individual graph. Graph-level anomaly detection targets anomalous individuals among various graphs. Not only the unique spatial structure and nodes/edges' attributes associated with each graph, but also the cross-graph structural and attribute patterns should be critically analyzed to identify these potential anomalies in the graph set (Ma et al., 2021).

Recent studies in deep graph representation have put great effort into encoding both the complex graph structural information and attribute information into vectors and then conducting graph analysis within the representation space (Wu et al., 2020). Although plenty of graph neural networks (GNNs) have been developed to learn expressive node representations via message passing

---

[1]The code is available at https://github.com/GmapAD/GmapAD

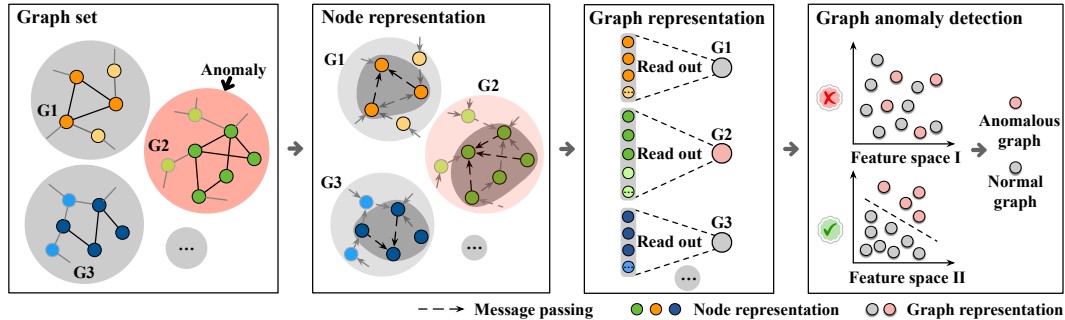

Figure 1: An overview of the learning feature space for graph-level anomaly detection. Each graph's representation is extracted from its own nodes' representations generated by message passing GNNs. Anomalies are then identified in the feature space. In a better feature space for anomaly detection (Feature Space II), anomalies ($\mathcal{G}_2$) and normal graphs ($\mathcal{G}_1$ and $\mathcal{G}_3$) should be well separated.

schema (Kipf & Welling, 2017; Veličković et al., 2018; Hamilton et al., 2017) and to read out the graph representation from nodes comprised in a single graph (Baek et al., 2021; Xu et al., 2019; Gallicchio & Micheli, 2020a), there remain significant challenges to directly applying existing GNNs for graph-level anomaly detection. (1) Most importantly, simply reading out graph representations using its own nodes cannot explicitly and fully capture the *inter*-graph information. For example, $\mathcal{G}_1$, $\mathcal{G}_2$, and $\mathcal{G}_3$'s representations shown in Figure 1 only maintain their *intra*-graph information while the rich cross-graph information is lost. This leads to unsatisfactory detection results and motivates us to design special read out functions to capture *intra*- and *inter*-graph information. (2) In the feature space, anomalies are also expected to locate away from normal graphs such that a clear boundary between them can be effectively learned such as Feature space II in Figure 1. (3) Lastly, the graph representation or the read out function should be interpretable. Noting that human understandable insights on the detected anomalies are vital for anomaly handling in real applications, but GNNs have been criticized for their low interpretability (Yuan et al., 2022; Pang et al., 2021).

To address the above mentioned challenges, in this work, we propose a novel graph mapping technique to learn effective representations for graph-level anomaly detection. Unlike the existing works that learn a graph's representation using its own nodes, our devised framework, **G**raph **map**ping **A**nomaly **D**etection (GmapAD), comprehensively explores both the complicated *intra*- and *inter*-graph structural and attribute information to map graphs into an interpretable latent space where anomalies and normal graphs are well separated. Specifically, GmapAD achieves a high degree of discriminativeness between anomalies and normal graphs by considering all nodes in the graph set and maps each single graph into the designed representation space according to the similarity between the graph and nodes.

Moreover, we notice that applying a simple graph mapping is non-trivial due to the massive number of nodes in the graph set and some nodes might contain non-valuable (or even misleading and defective) information for distinguishing anomalies, as validated in our experiment in Section 5.2. As a result, we further consider the informativeness of each node and propose a *differential evolutionary* algorithm to iteratively select the best-performing set of nodes for graph mapping. Eventually, anomalies and normal graphs are projected to different regions in the new feature space and can be distinguished effectively.

For validation, we conduct extensive experiments on nine real-world graph datasets by comparison with the state-of-the-arts using four commonly-used metrics, i.e., precision, recall, F1-scores and AUC. We also analyze the challenges and show the effectiveness of GmapAD modules through additional ablation tests. The results demonstrate that our proposed framework is superior to the existing works. In a nutshell, the main contributions of this work are as follows:

- To the best of our knowledge, this is the first graph-level anomaly detection framework that explores both the *intra*- and *inter*-graph information to find clues about anomalous graphs. The structural and attribute information/patterns within single graphs and cross-graphs can be effectively captured by our proposed GmapAD, which is also extendable to work jointly with the state-of-the-art graph neural networks.

- The graph mapping technique devised in this work is also explainable. Each graph's representation in the new feature space denotes its similarities with the most informative nodes chosen from the whole graph set, which is traceable, compared with black-box GNNs.

- The experiments on real-world datasets and ablation tests demonstrate the greater performance of GmapAD and validate our approach to the challenges.

## 2 RELATED WORK

**Graph neural networks.** To date, graph neural networks are generally implemented following the message-passing schema that aggregates neighborhoods' attributes for encoding the target node into a vector representation. These advanced neural networks, including the popular GCN (Kipf & Welling, 2017), GAT (Veličković et al., 2018) and GIN (Xu et al., 2019), have extensively explored the complex correlation between the spatial structure and attributes of a single graph and proposed different strategies to guide the information aggregation process. Although they have achieved promising results in various graph analysis tasks, such as link prediction (Wu et al., 2020) and node classification (Hamilton et al., 2017; Zhu et al., 2021), most existing works are limited to represent the whole graph using its own nodes through simple read out functions like mean or max pooling, resulting in sub-optimal solutions to graph-level tasks (Baek et al., 2021).

**Graph-level anomaly detection.** Graph-level anomaly detection is to spot anomalous graphs in a graph set. Different from node- (Dou et al., 2020; Tang et al., 2022; Bandyopadhyay et al., 2020) and edge-level (Yu et al., 2018; Duan et al., 2020) anomaly detection on a single graph, one should consider the complex structural and attribute information of each single graph (i.e., *intra*-graph information) as well as the cross-graph patterns (*inter*-graph information) for finding clues about graph-level anomalies. Thanks to the advancement of graph neural networks, recent graph-level anomaly detection techniques, such as OCGIN (Zhao & Akoglu, 2021) and OCGLT (Qiu et al., 2022), have applied GNNs to encode the *intra*-graph information into a vector and transferred graph-level anomaly detection to a conventional outlying data point detection problem in the representation space (Ma et al., 2021; Pang et al., 2021). Although these straightforward methods are convenient to apply, their capabilities in fusing the affluent *inter*-graph information is very limited and huge research gaps remain in this field. Nevertheless, these black-box GNN based methods are also criticized because they cannot provide human-understandable explanations to the learned representations and detected anomalies (Yuan et al., 2022).

**Differential evolutionary algorithm.** Differential evolution is a widely-used algorithm for finding optimal solutions to specific tasks based on random search (Storn & Price, 1997; Wu & Cai, 2014). Instead of performing brute-force search on all potential candidates, differential evolution algorithm adopts specially designed objective functions to guide the searching direction with guaranteed convergence (Hu et al., 2013; Vesterstrom & Thomsen, 2004; Rudolph, 1994). The key idea behind this algorithm is to simulate the natural evolution process, in which only the best candidates are retained while other candidates are updated through three operators: *crossover*, *mutation* and *selection*. Due to space limitation, we suggest interested readers to start with prior works like Ilonen et al. (2003); Zhang et al. (2016) and Qin et al. (2008) to better understand the evolutionary algorithm.

## 3 PRELIMINARIES AND PROBLEM DEFINITION

In this section, we provide the definitions of all related concepts used in the rest of the paper and the problem formulation of graph-level anomaly detection.

**Definition 1.** *Graph Set and Node Set.* Let $\mathbb{G} = (\mathcal{G}_1, \mathcal{G}_2, \ldots, \mathcal{G}_n)$ be a graph set containing $n$ individual attributed graphs and all nodes in the graph set form a node set $\mathbb{V} = \bigcup_i \mathbb{V}_i$, where $\mathbb{V}_i$ contains nodes in an individual graph $\mathcal{G}_i = (\boldsymbol{A}_i, \boldsymbol{X}_i)$ whose structure and node attributes are represented by the adjacency matrix $\boldsymbol{A}_i \in \{0, 1\}^{|\mathbb{V}_i| \times |\mathbb{V}_i|}$ and attribute matrix $\boldsymbol{X}_i$, respectively.

**Definition 2.** *Candidate and Candidate Pool.* We use candidate $\mathbb{C}_i \ni v_j, v_j \in \mathbb{V}$ to specifically denote a subset of nodes chosen from $\mathbb{V}$, and a candidate pool contains $p$ candidates, denoted as $\mathcal{C} = (\mathbb{C}_1, \mathbb{C}_2, \ldots, \mathbb{C}_p)$. The candidates and candidate pool are utilized to map individual graphs to new feature spaces. They are two key components of the differential evolutionary algorithm, to be detailed in Section 4.2.2.

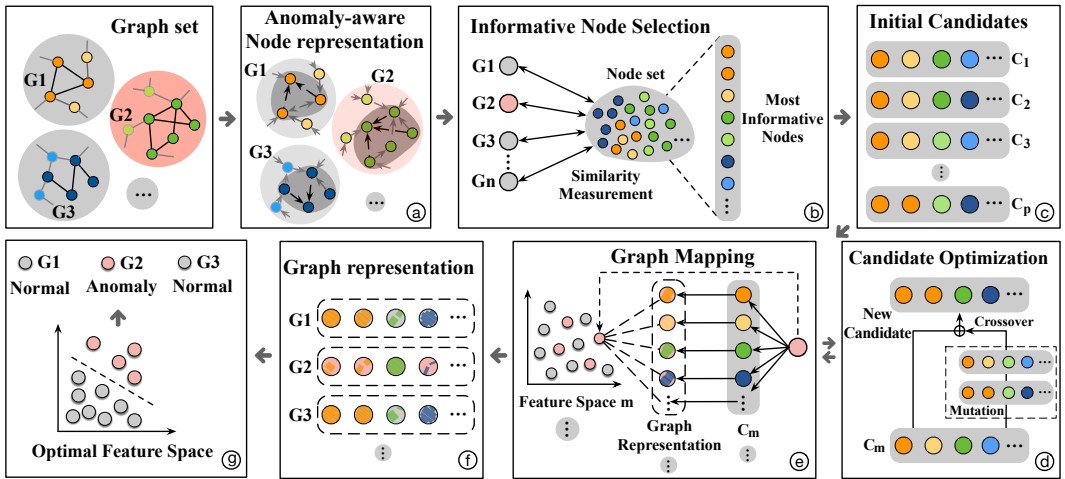

Figure 2: The GmapAD Framework. Given the graph set, ⓐ: GmapAD first encodes *intra*-graph information into anomaly-aware node representations. ⓑ: The most informative nodes are then selected from the node set based on the similarity between graphs and all nodes in the graph set. ⓒ: Initial candidates are chosen from the most informative nodes for graph mapping in step ⓔ. Based on the DE objective, candidates are optimized through ⓓ. Finally, each graph is mapped to the optimal feature space using the optimized candidate and anomalies are detected through ⓖ. In ⓕ, the more a graph is similar to a node in the candidate, the more area is covered by the node color in the corresponding dimension.

**Definition 3.** *Graph Mapping.* Given a candidate $\mathbb{C}_m$ in $\mathcal{C}$, graph mapping transforms each graph $\mathcal{G}_i$ into a new instance $\boldsymbol{h}_{\mathcal{G}_i}^m$ in the feature space. $\boldsymbol{h}_{\mathcal{G}_i}^m = (s(\mathcal{G}_i, v_1^m), \ldots, s(\mathcal{G}_i, v_{|\mathbb{C}_m|}^m))$, where $s(\mathcal{G}_i, v_k^m)$ denotes the similarity score of graph $\mathcal{G}_i$ and the $k$th node $v_k^m$ in $\mathbb{C}_m$. We use $\boldsymbol{h}_{\mathcal{G}_i}^m$ to specifically denote graph $\mathcal{G}_i$'s representation in the feature space that is built based on $\mathbb{C}_m$.

**Problem Definition.** Graph-level anomaly detection is defined to identify anomalous individual graphs in the given graph set $\mathbb{G}$. In this work, we learn graph representations in a semi-supervised manner. By this, the graph-level anomaly detection problem can be further defined as identifying anomalous graphs $\mathcal{G}_i$ using a small portion of graph labels $y_i = \{0, 1\}$, where 0 indicates that the graph is an anomaly.

## 4 GRAPH MAPPING ANOMALY DETECTION

In this section, we present the technical details of GmapAD, including anomaly-aware node representation learning in Section 4.1, graph mapping in Section 4.2.1, followed by the details of informative node selection and evolutionary candidate pool optimization in Section 4.2.2, and graph-level anomaly detection in Section 4.3.

### 4.1 ANOMALY-AWARE NODE REPRESENTATION

The spatial graph structure and node attributes encapsulate affluent information for identifying different classes of graphs. Although conventional message-passing graph neural networks have achieved success in encoding the *intra*-graph information into vector representation for facilitating node-level and graph-level analysis, they usually exhibit defective anomaly detection performance, especially on graph-level anomaly detection (Zhao & Akoglu, 2021; Ma et al., 2021; Qiu et al., 2022). One major reason is the extreme rarity of anomalies in the graph set since anomalies only take a very small proportion of the graph set (Noble & Cook, 2003; Akoglu et al., 2015). To learn more effective node and graph representations that better support graph-level anomaly detection, we first attempt to handle the imbalanced data distribution and learn anomaly-aware node representations such that the read out graph representations are distinguishable for anomaly detection.

For simplicity and extensibility, GmapAD employs the conventional message-passing based GNN (e.g., GCN, GAT) for learning node representations $\boldsymbol{z}_k^{\mathcal{G}_i, l}$ in $\mathcal{G}_i$ at each training iteration $l$ by ag-

gregating node $k$'s neighborhood information via $z_k^{\mathcal{G}_i,l} = \text{Agg}(\alpha_k z_k^{\mathcal{G}_i,l-1} + \sum_{j \in \mathbb{N}_{\mathcal{G}_i}(k)} \beta_j z_j^{\mathcal{G}_i,l-1})$, where $\text{Agg}(\cdot)$ is an aggregation function, such as sum or node pair-wise attentions. $\alpha_k$ and $\beta_j$ control the weights of $z_k^{\mathcal{G}_i,l-1}$ and $z_j^{\mathcal{G}_i,l-1}$ for learning $z_k^{\mathcal{G}_i,l}$, respectively. $\mathbb{N}_{\mathcal{G}_i}(k)$ denotes the neighbors of $k$ in graph $\mathcal{G}_i$ and $z_k^{\mathcal{G}_i,0}$ is node $k$'s attribute vector, a row vector in $\boldsymbol{X}_i$.

To ensure the learned node representations can preserve valuable *intra*-graph information for anomaly detection, we apply an anomaly-aware objective function to train the GNN by following Zhang et al. (2021). Specifically, given each individual training graph $\mathcal{G}_i$ and its label $y_i$, the representation of each node within the graph is learned via minimizing:

$$\mathcal{J} = \frac{1}{|\mathbb{G}_1|} \sum_{\mathcal{G}_i \in \mathbb{G}_1} \mathcal{L}[\phi(\frac{1}{|\mathbb{V}_i|} \sum_{k \in \mathcal{G}_i} z_k^{\mathcal{G}_i,l} \cdot \boldsymbol{W}), y_i] + \frac{1}{|\mathbb{G}_0|} \sum_{\mathcal{G}_j \in \mathbb{G}_0} \mathcal{L}[\phi(\frac{1}{|\mathbb{V}_j|} \sum_{k \in \mathcal{G}_j} z_k^{\mathcal{G}_j,l} \cdot \boldsymbol{W}), y_j], \quad (1)$$

where $\mathbb{G}_1$ and $\mathbb{G}_0$ contain normal and anomalous training graphs, respectively. $\mathcal{G}_i \in \mathbb{G}_1$ if $y_i = 1$, otherwise $\mathcal{G}_i \in \mathbb{G}_0$. $\phi(\cdot)$ predicts the probabilities of each graph $\mathcal{G}_i$ to be normal and anomalous. $\mathcal{L}(\cdot)$ measures the cross-entropy loss between the predicted graph labels and the ground truth. This class-wise training objective adaptively balances the weights of normal and anomalous training graphs for capturing the patterns of nodes within them.

## 4.2 GRAPH REPRESENTATION VIA DIFFERENTIAL EVOLUTIONARY GRAPH MAPPING

While the prior detailed node representation learning is capable to generate graph representations through special designed read out functions, such approaches are limited to extracting graph representations only considering *intra*-graph information, which inherently ignores valuable cross-graph information in the whole graph set and leads to sub-optimal solutions. Nevertheless, directly reading an individual graph's representation from node representations generated by graph neural networks also encounters severe low interpretability due to the black-box GNNs.

Driven by existing works in multi-instance learning and the feature mapping criteria between bags and instances (Wu et al., 2018b; Zhang et al., 2016), we repurpose the mapping criteria for graph representation learning by exploring the correlation between a single graph and a set of nodes. Specifically, given the generated anomaly-aware node representations, we first select the most informative $k$ nodes in $\mathbb{V}$ as a candidate, and then adopt a specially designed differential evolutionary algorithm (DE) to find the optimal graph mapping strategy for the eventual graph representation that will be used for distinguishing anomalies in the graph set as illustrated in steps ⓑ-ⓕ in Figure 2. We start with our proposed graph mapping method, followed by explanations on how we use DE to find the best performing candidate.

### 4.2.1 EXPLAINABLE GRAPH MAPPING

Graph mapping aims at encoding an individual graph $\mathcal{G}_i$ into a new instance $\boldsymbol{h}_{\mathcal{G}_i}^m \in \mathbb{R}^{|\mathbb{C}_m|}$ in the new feature space given a set of nodes in candidate $\mathbb{C}_m$. Notably, nodes in $\mathbb{C}_m$ are collected from different graphs, as shown in Figure 2. Each dimension in the new feature $\boldsymbol{h}_{\mathcal{G}_i}^m = (s(\mathcal{G}_i, v_1^m), \dots, s(\mathcal{G}_i, v_{|\mathbb{C}_m|}^m))$ denotes the proximity of graph $\mathcal{G}_i$ and a corresponding node $v_j^m$ in $\mathbb{C}_m$. For simplicity, in GmapAD, the similarity is measured as:

$$s(\mathcal{G}_i, v_j^m) = ||\frac{1}{|\mathbb{V}_i|} \sum_{k \in \mathcal{G}_i} z_k^{\mathcal{G}_i,l} - z_j^{\mathcal{G}_m,l}||, \quad (2)$$

where $z_k^{\mathcal{G}_i,l}$ denotes the learned representation of node $k$ in graph $\mathcal{G}_i$, while $z_j^{\mathcal{G}_m,l}$ is the representation of node $j$, which belongs to candidate $\mathbb{C}_m$. Through graph mapping, graphs can be clustered into different groups regarding their similarities to nodes in the candidate, which can be inherently utilized as a benchmark and normal graphs and anomalies will show divergent similarity patterns if the candidate is carefully selected.

Compared to conventional GNN-based models that generate graph-level representations through black-box neural networks, this graph mapping process is traceable and explainable. The value of each dimension in the graph representation explicitly denotes the proximity between the graph and a specific candidate node. Each candidate node, whilst, is ultimately selected from the graph set and can be traced back to its belonging graph, which is achieved by our specially designed DE algorithm detailed in the following section.

### 4.2.2 CANDIDATE SELECTION AND OPTIMIZATION

To facilitate better anomaly detection performance, anomalies and normal graphs are expected to locate far away in the feature space so that their boundaries are clear and easy to identify (e.g., Feature Space II in Figure 1). The chosen candidate for graph mapping is critical to gain such discriminativeness and we achieve this by first selecting the most informative nodes from all normal graphs and then find the best candidate for graph mapping through a specially designed DE algorithm.

**Informative node selection.** Since normal graphs are the majority in the graph set, instead of using all nodes in the node set $\mathbb{V}$ for graph mapping, we only select the top $k$ nodes that are most similar to normal graphs such that the computational and storage cost of graph mapping is minimized and the negative effect introduced by the curse of dimensionality (Bellman, 1966) can be alleviated. Practically, GmapAD selects the top $k$ nodes from normal graphs based on the cosine similarity, which measures a node $j$ and a normal graph $\mathcal{G}_i$'s similarity score as $\frac{\frac{1}{|\mathbb{V}_i|}\sum_{\epsilon\in\mathcal{G}_i}\boldsymbol{h}_\epsilon^{\mathcal{G}_i,l}\cdot\boldsymbol{h}_j^{\mathcal{G}_m,l}}{|\frac{1}{|\mathbb{V}_i|}\sum_{\epsilon\in\mathcal{G}_i}\boldsymbol{h}_\epsilon^{\mathcal{G}_i,l}|\cdot|\boldsymbol{h}_j^{\mathcal{G}_m,l}|}$, and for each node $j$, we take the sum of its similarity scores with all normal graphs as its final score. Eventually, the top $k$ nodes with the highest scores become the initial candidate $\mathbb{C}_{init}$ leveraged in the following DE algorithm.

**Differential evolutionary candidate optimization.** Given the initial candidate selected from the whole graph set, although each node in the candidate has high proximity to normal graphs, it is not guaranteed that conducting graph-node feature mapping using all of them will lead to the best detection performance (as validated in our ablation test in Section 5.2). Thus, we further establish a rule to optimize the candidate that all normal graphs should be similar in the new feature space while normal graphs and anomalies are dissimilar. Accordingly, due to DE's guaranteed convergence (Hu et al., 2013; Rudolph, 1994), we propose a novel objective function for guiding the DE algorithm and adaptively learn the best candidate $\mathbb{C}_{op}$ from different combinations of nodes in $\mathbb{C}_{init}$ through mutation, crossover, and selection (as shown in ⓓ in Figure 2). More specifically, given a candidate $\mathbb{C}_m$, our objective is to find the best diagonal node selection matrix $\boldsymbol{I}_{op}$ that maps graphs into a distinguishable space with the best anomaly detection performance, where $diag(\boldsymbol{I}_{op})$ is an indicator vector and $diag(\boldsymbol{I}_{op})_e = 1$ if $v_e \in \mathbb{C}_{op}$, otherwise 0. We obtain $\boldsymbol{I}_{op}$ following:

**(1) Initialization.** In this stage, we randomly initialize $p$ different candidate selection matrices $\mathbb{I}^0 = [\boldsymbol{I}_1^0, \ldots, \boldsymbol{I}_p^0]$ for generating $p$ divergent candidates that forms a set $\mathcal{C}^0 = [\mathbb{C}_1^0, \ldots, \mathbb{C}_p^0]$. Specifically, for candidate selection matrix $\boldsymbol{I}_i^0$, each entry $e$ in its indicator vector $diag(\boldsymbol{I}_i^0) \in \{0,1\}^{|\mathbb{C}_{init}|}$ is drawn from a Bernoulli distribution $diag(\boldsymbol{I}_i^0)_e \sim \mathcal{B}(p_s)$. This intuitive binary selection process directly chooses nodes from the graph set and the selected node can be easily traced back to its belonging graph.

**(2) Evaluation.** Given the candidates in $\mathcal{C}^t$, where $0 \le t \le et$ denotes the evolution iteration, we then map all graphs to $p$ feature spaces following the graph mapping detailed in Section 4.2.1 using each candidate $\mathbb{C}_i^t$ in $\mathcal{C}^t$ and apply a binary SVM (Chang & Lin, 2011) to classify graphs as anomalies or normal in each space. A performance score is assigned to each candidate according to the anomaly detection performance, which is quantified as:

$$score(\mathbb{C}_i^t) = \frac{1}{|\mathbb{G}_{train}|} \sum_{\mathcal{G}_k \in \mathbb{G}_{train}} \mathcal{L}(y_{\mathcal{G}_k}, \hat{y}_{\mathcal{G}_k}), \tag{3}$$

and

$$\mathcal{L}(y_{\mathcal{G}_k}, \hat{y}_{\mathcal{G}_k}) = \begin{cases} max(0, 1 - y_{\mathcal{G}_k} \cdot \hat{y}_{\mathcal{G}_k}), & y_{\mathcal{G}_k} = 1 \\ max(0, \hat{y}_{\mathcal{G}_k}), & y_{\mathcal{G}_k} = 0 \end{cases} \tag{4}$$

where $y_k$ and $\hat{y}_k$ are the ground truth and predicted label of a training graph $\mathcal{G}_k$, respectively.

**(3) Updating.** Given the score of each candidate, we then generate an updated set of candidates $\mathbb{C}^{t+1}$ based on $\mathbb{C}^t$ through mutation and crossover. In the mutation process, a new candidate prototype is created using candidate selection matrix $\boldsymbol{I}_{new}^{t+1}$, which is created as:

$$diag(\boldsymbol{I}_{new}^{t+1})_e = \begin{cases} 1, & d \ge 2 \\ 0, & d < 2 \end{cases} \tag{5}$$

where $d = diag(\boldsymbol{I}_{r1}^t)_e + \mu(diag(\boldsymbol{I}_{r2}^t)_e + diag(\boldsymbol{I}_{r3}^t)_e + diag(\boldsymbol{I}_{r4}^t)_e + diag(\boldsymbol{I}_{r5}^t)_e)$, $\mu$ is the predefined mutation rate, and $r1, r2, r3, r4,$ and $r5$ are divergent integers randomly selected between $[1, p]$ to

maintain the diversity of candidates, as specified in existing works (Ahmad et al., 2021; Deng et al., 2021; Zhang et al., 2016). We set the threshold of $d$ as 2 to ensure $diag(\boldsymbol{I}_{new}^{t+1})_e$ has equal possibilities to be 1 or 0 in our setting $\mu = 0.5$. A binominal crossover is then applied to get the final candidate selection matrix $\boldsymbol{I}_{new}^{t+1}$ as:

$$diag(\boldsymbol{I}_{new}^{t+1})_e = \begin{cases} diag(\boldsymbol{I}_{new}^{t+1})_e, & p(r) < \text{CR or } r = e \\ diag(\boldsymbol{I}_{new}^{t})_e, & otherwise \end{cases} \tag{6}$$

where $p(r)$ is a random number drawn in range $[0, 1]$ following the uniform distribution, CR is a predefined crossover rate, $r$ is a random integer between $[1, |\mathbb{C}_i^t|]$. Once the new candidate is generated, we quantify its score following the prior evaluation step and replace a less performing candidate $\mathbb{C}_i^t$ by $\mathbb{C}_i^{t+1}$ since $score(\mathbb{C}_i^{t+1}) < score(\mathbb{C}_i^t)$.

This evolutionary candidate optimization is applied to find the most supportive candidate within $et$ iterations. Finally, $\mathbb{C}_{op}$ is chosen for mapping all graphs into a new feature space in which anomalies and normal graphs can be best separated. And, a graph $\mathcal{G}_i$'s representation can be generated using $\mathbb{C}_{op}$ with candidate selection matrix $\boldsymbol{I}_{op}$ as the following:

$$\boldsymbol{h}_{\mathcal{G}_i} = [s(\mathcal{G}_i, v_1^{op} \cdot diag(\boldsymbol{I}_{op})_1), \ldots, s(\mathcal{G}_i, v_{|\mathbb{C}_{op}|}^{op} \cdot diag(\boldsymbol{I}_{op})_{|\mathbb{C}_{op}|})) \tag{7}$$

### 4.3 GRAPH-LEVEL ANOMALY DETECTION

After the last evolution iteration of DE, GmapAD eventually learns the best candidate and a SVM classifier for assigning graph labels. Given test graphs in the input set, the model maps each graph into the new feature space and labels them regarding their locations in the space.

## 5 EXPERIMENTS

For validating the performance and key modules of GmapAD, we 1) conduct extensive experiments for detecting anomalies on nine commonly used graph datasets by comparison with the state-of-the-arts, and 2) perform a thorough ablation test on the devised model.

### 5.1 ANOMALY DETECTION PERFORMANCE

#### 5.1.1 DATASETS

The nine benchmark datasets include two mostly-used brain networks datasets[2] (i.e., KKI and OHSU), and seven repurposed binary graph classification datasets published on site[3]. Specifically, AIDS, MUTAG, Mutagenicity, NCI1 and PROTEINS are collected from biochemistry while IMDB-BINARY and REDDIT-BINARY are from online social networks (Cai & Wang, 2018; Wu et al., 2018a; Gallicchio & Micheli, 2020b;a). The KKI and OHSU brain network datasets are constructed from the functional magnetic resonance image (fMRI) atlas of the whole brain for brain disorder analysis (Van Den Heuvel & Pol, 2010; Pan et al., 2016; Hernández-Pérez et al., 2021), and since their labels indicate abnormal brain disorders, we use them directly without any further processing. For the 7 graph classification datasets, we follow previous works in (Zhao & Akoglu, 2021) and (Qiu et al., 2022) and repurposed them for graph-level anomaly detection by downsampling the deviating class. The detailed descriptions and statistics of these datasets are given in Appendix A.

#### 5.1.2 BASELINES

We compare our devised framework GmapAD with six state-of-the-art GNN models, GCN (Kipf & Welling, 2017), GAT (Veličković et al., 2018), g-U-Nets (Gao & Ji, 2019), SAGPool (Lee et al., 2019), DIFFPOOL (Ying et al., 2018), and GMT Baek et al. (2021), and two graph-level anomaly detection methods, OCGIN (Zhao & Akoglu, 2021) and OCGTL (Qiu et al., 2022). For a fair comparison, we add the anomaly-aware loss function proposed in Section 4.1 to the six GNN models for reassigning them for graph anomaly detection and use GCN and GAT as the basis of GmapAD for graph mapping. g-U-Nets, SAGPool, DIFFPOOL and GMT adopt different pooling strategies

---

[2]https://github.com/GRAND-Lab/graph_datasets
[3]https://chrsmrrs.github.io/datasets/

Table 1: Detection precision scores on nine datasets. (Best in bold)

| Dataset | GCN | GAT | g-U-Nets | DiffPool | SAGPool | GMT | OCGIN | OCGLT | GmapAD-GCN | GmapAD-GAT |
|---------|-----|-----|----------|----------|---------|-----|-------|-------|------------|------------|
| KKI | 0.53±0.3 | 0.54±0.3 | 0.38±0.2 | 0.46±0.1 | 0.36±0.2 | 0.46±0.2 | 0.40±0.2 | 0.43±0.2 | 0.63±0.3 | **0.67±0.4** |
| OHSU | 0.58±0.3 | 0.55±0.2 | 0.51±0.3 | 0.48±0.2 | 0.46±0.2 | 0.43±0.2 | 0.57±0.1 | 0.64±0.2 | 0.59±0.1 | **0.66±0.2** |
| AIDS | 0.97±0.1 | 0.97±0.1 | 0.97±0.1 | 0.97±0.1 | 0.92±0.1 | **0.98±0.3** | 0.96±0.1 | 0.96±0.2 | **0.98±0.1** | **0.98±0.1** |
| MUTAG | 0.83±0.2 | 0.77±0.3 | 0.88±0.1 | 0.75±0.1 | 0.80±0.1 | 0.79±0.1 | 0.59±0.1 | 0.72±0.1 | **0.93±0.3** | 0.84±0.2 |
| Mutagenicity | 0.87±0.1 | 0.88±0.1 | 0.88±0.1 | 0.72±0.1 | 0.88±0.1 | 0.90±0.1 | 0.89±0.2 | 0.88±0.1 | 0.90±0.1 | **0.91±0.1** |
| NCI1 | 0.88±0.3 | 0.87±0.2 | 0.89±0.1 | 0.61±0.1 | 0.91±0.1 | 0.91±0.1 | 0.90±0.1 | 0.92±0.1 | **0.93±0.2** | **0.93±0.3** |
| PROTEINS | 0.87±0.1 | 0.85±0.2 | 0.87±0.3 | 0.75±0.1 | 0.87±0.1 | 0.88±0.1 | 0.85±0.1 | 0.86±0.2 | **0.89±0.1** | 0.87±0.2 |
| IMDB-BINARY | 0.91±0.1 | 0.87±0.3 | 0.94±0.1 | 0.76±0.1 | 0.90±0.1 | 0.93±0.1 | 0.93±0.1 | 0.88±0.1 | **0.98±0.1** | 0.92±0.1 |
| REDDIT-BINARY | 0.88±0.1 | 0.85±0.1 | 0.89±0.1 | 0.74±0.1 | 0.91±0.1 | **0.93±0.2** | 0.83±0.1 | 0.89±0.1 | **0.93±0.1** | 0.91±0.1 |

and specially designed pooling layers for learning graph-level representations. Both baseline graph anomaly detection models, OCGIN and OCGLT, apply a one-class SVDD classifier to learn a hypersphere using only normal graphs, and anomalies are distinguished as those lying outside the hypersphere. Open-sourced implementations of these baselines are provided in Appendix B.

### 5.1.3 EXPERIMENTAL SETTINGS

In our experiment, we denote GmapAD-GCN and GmapAD-GAT as variants of our model using GCN and GAT as basis, respectively. We use Adam (Kingma & Ba, 2014) (learning rate is 0.005, weight decay is 0.0005) as the optimizer for fine-tuning GNN parameters. In GmapAD-GCN, we stack 2 GCN layers, while in GmapAD-GAT, we use 2 GAT layers, each of which uses 8 heads. The dimensions of GNN layers are set to 128 and 64, respectively, and the dimension of $W$ in Equation 1 is 64. For the evolutionary graph-node feature mapping, we set $k$ to 64 when selecting the top-$k$ most informative nodes and set the size of candidate pool to 30. For DE, we set the possibility $p_s$ of the Bernoulli distribution as 0.5, mutate rate as 0.5, crossover rate as 0.9 and iterations for evolution as 2,000 (following previous DE works (Zhang et al., 2016; Wu & Cai, 2014)). Each dataset is randomly shuffled and split for training, validation and test with ratios of 80%, 10%, and 10%. For the baselines, we use their published settings unless the parameters are specially identified in the original paper. All the experiments are conducted on a Rocky Linux 8.6 (Green Obsidian) server with a 12-core CPU, 1 Nvidia V100 GPU and 60Gb RAM. The average and standard deviation of 10-fold test results are reported. The additional parameter sensitivity test results are provided in Appendix D.

### 5.1.4 RESULT ANALYSIS

From the results in Tables 1, 2, and 3, the two variants of GmapAD, namely GmapAD-GCN and GmapAD-GAT, achieve better results compared to other baselines on all datasets regarding precision, recall and F1-scores, because all baselines only consider *intra*-graph information for detecting anomalies in the graph set, which inherently indicates that one should explore the *inter*-graph information more comprehensively for generating graph representations so that potential graph-level anomalies can be more effectively identified. The conventional GCN and GAT models we utilize for validation can also achieve comparative results to other state-of-the-art on graph-level anomaly detection when trained with the anomaly-aware loss detailed in Section 4.1. g-U-Nets, SAGPool, DIFFPOOL and GMT obtain the second-best results which can be attributed to their specially designed pooling methods for reading graph-level representations using nodes within the graph, while GmapAD explores the most informative nodes considering the whole graph set. We can also see that there remain significant gaps to improve the detection performance on the two brain networks and the low results of all methods on them can be reasoned to the relatively smaller size of datasets. Nevertheless, the higher recall scores of all methods compared to their precision and F1-scores imply the methods can capture most of the true anomalies while the number of predicted false positives by different models is relatively high. Both problems can be future research in this direction. Due to space limitation, AUC results are reported in Appendix C.

### 5.2 ABLATION TESTS

Apart from anomaly detection performance validation, we perform further ablation tests to validate our approach to the challenges pinpointed in Section 1. Since GmapAD's training process comprises two main stages, i.e., anomaly-aware node representation learning and evolutionary graph mapping, we report GNN basis' detection performance to show the effectiveness of the learned node representations and then validate the differential evolutionary graph mapping method by conducting graph

Table 2: Detection recall scores on nine datasets. (Best in bold)

| Dataset | GCN | GAT | g-U-Nets | DiffPool | SAGPool | GMT | OCGIN | OCGLT | GmapAD-GCN | GmapAD-GAT |
|---|---|---|---|---|---|---|---|---|---|---|
| KKI | 0.35±0.1 | 0.62±0.4 | 0.32±0.1 | 0.46±0.1 | 0.68±0.5 | 0.83±0.4 | 0.38±0.2 | 0.42±0.2 | **0.83±0.3** | 0.80±0.4 |
| OHSU | 0.70±0.2 | 0.49±0.2 | 0.45±0.4 | 0.49±0.2 | 0.65±0.2 | 0.68±0.4 | 0.57±0.1 | 0.62±0.2 | **0.82±0.2** | 0.64±0.2 |
| AIDS | 0.58±0.1 | 0.72±0.1 | 0.73±0.2 | 0.93±0.1 | 0.82±0.1 | **1.0±0.0** | 0.49±0.1 | 0.97±0.1 | 0.99±0.1 | 0.99±0.1 |
| MUTAG | 0.82±0.2 | 0.61±0.1 | 0.85±0.1 | 0.80±0.1 | 0.82±0.1 | 0.76±0.1 | 0.36±0.1 | 0.73±0.1 | **0.95±0.3** | 0.89±0.2 |
| Mutagenicity | 0.71±0.1 | 0.79±0.1 | 0.95±0.2 | 0.74±0.1 | 0.95±0.1 | 0.95±0.1 | 0.49±0.1 | 0.88±0.1 | **0.98±0.1** | **0.98±0.1** |
| NCI1 | 0.75±0.3 | 0.78±0.1 | 0.96±0.1 | 0.58±0.1 | 0.95±0.1 | 0.98±0.1 | 0.48±0.1 | 0.92±0.1 | **0.99±0.1** | **0.99±0.1** |
| PROTEINS | 0.57±0.1 | 0.64±0.4 | 0.71±0.1 | 0.68±0.1 | 0.92±0.1 | 0.95±0.1 | 0.48±0.1 | 0.87±0.1 | **0.99±0.1** | 0.97±0.2 |
| IMDB-BINARY | 0.70±0.1 | 0.85±0.3 | 0.98±0.1 | 0.77±0.1 | 0.98±0.1 | 0.95±0.1 | 0.98±0.1 | 0.89±0.1 | **0.99±0.1** | 0.94±0.1 |
| REDDIT-BINARY | 0.83±0.1 | 0.92±0.1 | 0.78±0.2 | 0.68±0.1 | 0.91±0.1 | 0.93±0.1 | 0.45±0.1 | 0.89±0.1 | 0.94±0.1 | **0.96±0.1** |

Table 3: Detection F1 scores on nine datasets. (Best in bold)

| Dataset | GCN | GAT | g-U-Nets | DiffPool | SAGPool | GMT | OCGIN | OCGLT | GmapAD-GCN | GmapAD-GAT |
|---|---|---|---|---|---|---|---|---|---|---|
| KKI | 0.36±0.2 | 0.56±0.3 | 0.34±0.1 | 0.44±0.1 | 0.46±0.3 | 0.58±0.3 | 0.38±0.2 | 0.43±0.2 | 0.71±0.3 | **0.73±0.3** |
| OHSU | 0.59±0.2 | 0.51±0.2 | 0.36±0.3 | 0.47±0.2 | 0.51±0.2 | 0.51±0.3 | 0.57±0.1 | 0.61±0.2 | **0.68±0.7** | 0.63±0.2 |
| AIDS | 0.73±0.1 | 0.83±0.1 | 0.83±0.1 | 0.98±0.1 | 0.86±0.1 | **1.0±0.0** | 0.65±0.1 | 0.96±0.1 | 0.99±0.1 | 0.99±0.1 |
| MUTAG | 0.86±0.1 | 0.75±0.1 | 0.82±0.1 | 0.83±0.1 | 0.81±0.1 | 0.78±0.1 | 0.44±0.1 | 0.71±0.2 | **0.96±0.3** | 0.86±0.1 |
| Mutagenicity | 0.81±0.1 | 0.87±0.1 | 0.92±0.1 | 0.72±0.1 | 0.91±0.1 | 0.92±0.1 | 0.63±0.1 | 0.87±0.2 | **0.94±0.1** | **0.94±0.1** |
| NCI1 | 0.84±0.3 | 0.85±0.1 | 0.94±0.1 | 0.59±0.1 | 0.95±0.1 | 0.94±0.1 | 0.63±0.1 | 0.92±0.1 | 0.96±0.2 | **0.97±0.2** |
| PROTEINS | 0.71±0.1 | 0.73±0.3 | 0.78±0.1 | 0.71±0.1 | 0.92±0.1 | 0.91±0.1 | 0.61±0.1 | 0.85±0.3 | **0.94±0.1** | 0.93±0.2 |
| IMDB-BINARY | 0.81±0.1 | 0.88±0.2 | 0.94±0.1 | 0.91±0.2 | 0.94±0.1 | **0.95±0.1** | 0.61±0.1 | 0.87±0.2 | **0.95±0.1** | 0.94±0.1 |
| REDDIT-BINARY | 0.88±0.1 | 0.92±0.1 | 0.82±0.1 | 0.69±0.1 | 0.93±0.1 | 0.93±0.2 | 0.58±0.1 | 0.88±0.1 | 0.94±0.1 | **0.95±0.1** |

mapping using all top-$k$ informative nodes in the whole graph set without further optimization (denoted as + Full mapping), followed by graph mapping using the best candidate selected through differential evolution (denoted as +DE).

**Graph mapping performance.** As illustrated in Tables 1, 2, 3, and 4, GmapAD-GCN's superior detection performance mainly comes from our proposed differential evolutionary graph mapping method, improved more than 10% and 2% on the GCN basis and full mapping, respectively, on the three datasets. By comparing GCN and full mapping, we find that the straightforward graph mapping using all top-$k$

Table 4: Ablation test (Best in bold).

| Dataset | Variants | F1-score | Precision | Recall |
|---|---|---|---|---|
| KKI | GCN | 0.36±0.2 | 0.53±0.3 | 0.80±0.3 |
| | + Full mapping | 0.43±0.3 | 0.40±0.5 | 0.74±0.4 |
| | + DE | **0.71±0.3** | **0.63±0.3** | **0.83±0.3** |
| MUTAG | GCN | 0.86±0.1 | 0.83±0.2 | 0.82±0.2 |
| | + Full mapping | 0.91±0.1 | 0.91±0.2 | 0.91±0.3 |
| | + DE | **0.96±0.3** | **0.93±0.3** | **0.95±0.3** |
| Mutagenicity | GCN | 0.81±0.1 | 0.87±0.1 | 0.71±0.1 |
| | + Full mapping | 0.93±0.1 | 0.84±0.1 | **0.99±0.1** |
| | + DE | **0.94±0.1** | **0.90±0.1** | 0.98±0.1 |

informative nodes cannot guarantee a satisfying detection performance. For example, GCN outperforms full mapping regarding the precision and recall scores on KKI dataset. This is because some nodes might contain non-informative or even defective information that may blur the boundary between anomalies and normal graphs, as prior mentioned in Section 4.2.1.

**Differential evolutionary algorithm's performance.** +DE's better performance over full mapping demonstrates that our devised DE algorithm can further improve the detection performance by iteratively selecting the best candidate nodes for graph mapping while maintaining the traceability of graph mapping for representation explanation.

## 6 CONCLUSION

In this paper, we have devised a novel graph-level anomaly detection framework based on graph representations learned through deep evolutionary graph mapping. From our study, we pinpoint that investigating both the abundant *inter-* and *intra-*graph information in the graph set is mandatory for capturing the deviating patterns of anomalies, which however has not yet been sufficiently studied in existing works. To bridge the gap, our developed framework, GmapAD, first utilizes a GNN to encapsulate the spatial graph structure and attributes into node representations and then iteratively selects the best subset of informative nodes in the graph set for mapping graphs into the new feature space (according to the similarity between each graph and nodes in the subset). Eventually, anomalies and normal graphs will be mapped to the deviating locations in the feature space and thus can be classified effectively using conventional machine learning techniques such as SVM. The extensive experiments on nine widely-used real-world datasets show that GmapAD outperforms the five state-of-the-art baselines, and our approach to the challenges discussed early in the paper is validated through the additional ablation tests.

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

## A    DOWNSAMPLED DATASET DESCRIPTION

Statistics of the nine raw datasets, including the number of graphs in each class, the average number of nodes and edges are given in Table 5. For the KKI and OHSU datasets, $\mathcal{G}_0$ denotes brain networks/graphs with disorders while $\mathcal{G}_1$ denotes the normal class. For the rest seven datasets, we follow previous works in Zhao & Akoglu (2021); Qiu et al. (2022) and downsample $\mathcal{G}_0$ as anomalies (only 10% data samples are kept). The meanings of $\mathcal{G}_0$ and $\mathcal{G}_1$ are also summarized in Table 6.

Table 5: Dataset statistics

| Dataset | KKI | | OHSU | | AIDS | | MUTAG | | Mutagenicity | | PROTEINS | | NCI1 | | IMDB | | REDDIT | |
|---|---|---|---|---|---|---|---|---|---|---|---|---|---|---|---|---|---|---|
| | $\mathcal{G}_0$ | $\mathcal{G}_1$ | $\mathcal{G}_0$ | $\mathcal{G}_1$ | $\mathcal{G}_0$ | $\mathcal{G}_1$ | $\mathcal{G}_0$ | $\mathcal{G}_1$ | $\mathcal{G}_0$ | $\mathcal{G}_1$ | $\mathcal{G}_0$ | $\mathcal{G}_1$ | $\mathcal{G}_0$ | $\mathcal{G}_1$ | $\mathcal{G}_0$ | $\mathcal{G}_1$ | $\mathcal{G}_0$ | $\mathcal{G}_1$ |
| #$\mathcal{G}$ | 37 | 46 | 35 | 44 | 400 | 1600 | 63 | 125 | 2401 | 1936 | 663 | 450 | 2053 | 2057 | 500 | 500 | 1000 | 1000 |
| avg.#V | 190.0 | 190.0 | 190.0 | 190.0 | 37.61 | 10.2 | 13.9 | 14.9 | 29.4 | 31.5 | 49.9 | 22.9 | 25.7 | 34.1 | 20.1 | 19.4 | 641.3 | 218.0 |
| avg.#E | 237.4 | 239.3 | 400.5 | 381.1 | 80.9 | 20.4 | 29.2 | 44.8 | 60.6 | 62.7 | 188.1 | 83.1 | 55.3 | 73.9 | 193.6 | 192.6 | 1471.9 | 519.1 |

## B    BASELINE IMPLEMENTATIONS

In our experiment, we strictly follow the official implementations of the baselines and their corresponding code repository are as follows:

g-U-Nets. `https://github.com/HongyangGao/Graph-U-Nets`.

SAGPool. `https://github.com/inyeoplee77/SAGPool`.

DIFFPOOL. `https://github.com/RexYing/diffpool`.

GMT. `https://github.com/JinheonBaek/GMT`.

OCGIN. `https://github.com/LingxiaoShawn/GLOD-Issues`.

OCGLT. `https://github.com/boschresearch/GraphLevel-AnomalyDetection`.

## C    ADDITIONAL EXPERIMENTAL RESULTS

We provide experimental results regarding the ROC-AUC score here. Specifically, as our proposed GmapAD learns an SVM classifier and directly assigns labels for graphs as anomalies or normal, we use the distance between each graph representation and the margin learned by SVM as its score. As can be seen in Table 7, GmapAD achieves the best performance in most cases.

## D    PARAMETER SENSITIVITY

As detailed in Section 4 and the experimental settings in Section 5.1.3, GmapAD's main hyper-parameters are: 1) dimensions of the GNN layers, 2) $k$ value used for selecting the top-$k$ most informative nodes in the graph set, and 3) mutation rate and crossover rate in the differential evolutionary algorithm. For validation purposes and because GmapAD is two-staged, we test GmapAD's sensitivity to GNN parameters and $k$ value separately. We set $k$ as 64 when testing GNN dimensions' impacts. Similarly, we set the GNN dimensions as 128 and 64 when testing different $k$ values. To be noticed, for the DE algorithm-related parameters, we follow previous works in Zhang et al. (2016) and Wu & Cai (2014) since both rates have already been validated. All tests are conducted on a representative dataset – MUTAG and GmapAD's detection performance regarding precision, recall and F1 scores are reported.

### D.1    GNN PARAMETER SENSITIVITY

In this experiment, we use GCN as the basis of GmapAD and conduct a grid search on the dimensions of the two GCN layers, where each layer's dimension is set as $[32, 64, 128, 256]$. As can be seen from Figure 3, the settings of GCN dimensions have little impact on GmapAD-GCN's performance. This maybe because the graph mapping takes the similarities between graph and candidate

Table 6: Graph label description

| Dataset | Class | Description |
|---|---|---|
| KKI | $\mathcal{G}_0$ | Brain networks with attention deficit hyperactivity disorder |
| | $\mathcal{G}_1$ | Health brain networks |
| OHSU | $\mathcal{G}_0$ | Brain networks with hyperactive-impulsive disorder |
| | $\mathcal{G}_1$ | Health brain networks |
| AIDS | $\mathcal{G}_0$ | Chemical compounds inactive against HIV |
| | $\mathcal{G}_1$ | Chemical compounds active against HIV |
| MUTAG | $\mathcal{G}_0$ | Nitroaromatic compounds mutagenicity on Salmonella typhimurium |
| | $\mathcal{G}_1$ | Nitroaromatic compounds non-mutagenicity on Salmonella typhimurium |
| Mutagenicity | $\mathcal{G}_0$ | Chemical compounds categorized as mutagen drugs |
| | $\mathcal{G}_1$ | Chemical compounds categorized as non-mutagen |
| PROTEINS | $\mathcal{G}_0$ | Enzymes proteins |
| | $\mathcal{G}_1$ | Non-enzymes proteins |
| NCI1 | $\mathcal{G}_0$ | Chemical compounds active for non-small cell lung cancer |
| | $\mathcal{G}_1$ | Chemical compounds inactive for non-small cell lung cancer |
| IMDB-BINARY | $\mathcal{G}_0$ | Movie collaboration network extracted from romance movies |
| | $\mathcal{G}_1$ | Movie collaboration network extracted from action movies |
| REDDIT-BINARY | $\mathcal{G}_0$ | Discussion-based communities extracted from Reddit |
| | $\mathcal{G}_1$ | Question/answer-based communities extracted from Reddit |

Table 7: Detection AUC scores on nine datasets. (Best in bold)

| Dataset | GCN | GAT | g-U-Nets | DiffPool | SAGPool | GMT | OCGIN | OCGLT | GmapAD-GCN | GmapAD-GAT |
|---|---|---|---|---|---|---|---|---|---|---|
| KKI | 0.52±0.1 | 0.44±0.2 | 0.35±0.2 | 0.50±0.2 | 0.50±0.1 | 0.52±0.1 | 0.44±0.1 | 0.35±0.1 | **0.64±0.2** | 0.55±0.2 |
| OHSU | 0.62±0.2 | 0.51±0.1 | 0.53±0.1 | 0.59±0.2 | 0.53±0.2 | 0.57±0.2 | 0.50±0.1 | 0.46±0.2 | **0.69±0.2** | 0.64±0.3 |
| AIDS | 0.64±0.1 | 0.66±0.1 | **0.98±0.1** | 0.98±0.1 | 0.93±0.1 | **0.98±0.1** | 0.96±0.1 | 0.97±0.1 | 0.93±0.1 | 0.92±0.1 |
| MUTAG | 0.54±0.1 | 0.50±0.1 | 0.76±0.1 | 0.76±0.1 | 0.73±0.1 | 0.72±0.1 | 0.63±0.4 | 0.66±0.1 | **0.79±0.1** | 0.73±0.2 |
| Mutagenicity | 0.46±0.1 | 0.65±0.1 | 0.68±0.1 | 0.62±0.1 | 0.68±0.1 | **0.69±0.1** | 0.47±0.1 | 0.55±0.1 | **0.69±0.1** | 0.67±0.1 |
| NCI1 | 0.62±0.1 | 0.62±0.1 | 0.65±0.1 | 0.66±0.1 | 0.66±0.1 | **0.67±0.1** | 0.66±0.1 | 0.63±0.1 | **0.67±0.1** | **0.67±0.1** |
| PROTEINS | 0.65±0.1 | 0.61±0.1 | 0.83±0.1 | 0.73±0.1 | **0.88±0.1** | 0.66±0.1 | 0.39±0.1 | 0.69±0.1 | **0.88±0.1** | 0.79±0.2 |
| IMDB-BINARY | 0.67±0.1 | 0.69±0.1 | 0.76±0.1 | 0.69±0.1 | 0.85±0.1 | 0.77±0.1 | 0.61±0.3 | 0.67±0.3 | **0.87±0.1** | 0.81±0.1 |
| REDDIT-BINARY | 0.73±0.1 | 0.68±0.1 | 0.74±0.1 | 0.74±0.1 | 0.75±0.1 | **0.78±0.1** | 0.42±0.4 | 0.75±0.2 | 0.76±0.1 | 0.73±0.1 |

nodes for generating graph representations. Although the dimension of GNN varies, the similarities among graphs and the selected candidate nodes are rarely affected by it.

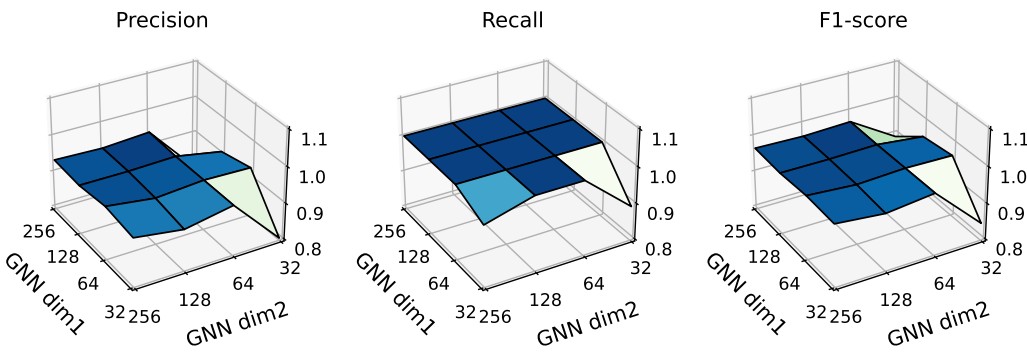

Figure 3: GNN parameter sensitivity.

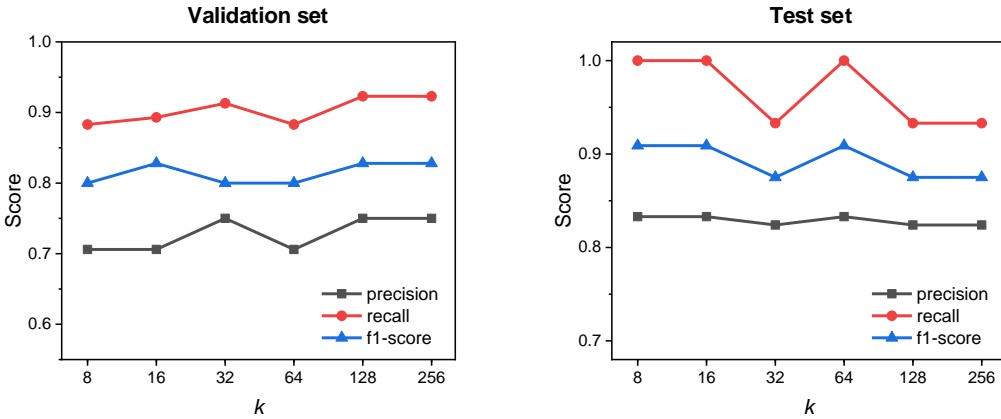

Figure 4: $k$ value sensitivity.

## D.2 $k$ VALUE SENSITIVITY

The value of $k$ determines the number of the most informative nodes for graph mapping and optimization. To validate $k$'s impact on GmapAD, we use GmapAD-GCN with fixed dimensions $(128, 64)$ and report the detection performance on validation and test sets under settings $k = [8, 16, 32, 64, 128, 256]$ in Figure 4.

We can observe slight changes in the precision, recall and F1-score under different $k$ values. This implies GmapAD can eventually find the best candidate from the given graph set for graph mapping and the best $k$ for MUTAG is $64$. Meanwhile, since a high value of $k$ will result in high dimensional graph representations, which introduces more computation cost, a balance between $k$ and the performance should be made when applied to real applications.

