# OpenReview forum: "Towards graph-level anomaly detection via deep evolutionary mapping"
_ICLR.cc/2023/Conference — Submitted to ICLR 2023_

### Official Review · Reviewer_BQDF · 2022-10-23

**Confidence:** 4
**Clarity, Quality, Novelty And Reproducibility:** The clarity can be improved. The nove…
**Correctness:** 3
**Technical Novelty And Significance:** 3
**Empirical Novelty And Significance:** Not applicable
**Recommendation:** 5

**Strength And Weaknesses:**

Strengths:
1. The essential idea of the article is to select some best candidate nodes for graph mapping, thus leveraging inter-graph information. The author proposed a novel differential evolutionary algorithm to determine the most supportive candidates.
2. The numerical outcomes are good, especially F1 score.
3. The ablation test shows the differential evolutionary process is essential after selecting k informative nodes as candidates.

Weakness:
1. There are some confusing notations in Section 4.2.2. For inforamtive node selection, the author states the top k nodes most similar to normal graph, but only gives the similarity calculation between a node j and a graph G_i , with no clue how node score is aggregated (by summation?) (2) For candidate initialization, rigorously speaking, it should be $\{0, 1\}^{| C_i |}$ not $\{0, 1\}^{| C_p |\}$; the parameter $p_s$ in Bernoulli distribution is unstated in the experiment setting. For updating a set of candidates, the subscript of diagonal matrix $I^t$ is supposed to range from 1 to p, according to the (1) Initialization. However, the author claims to select r1, r2, r3, r4, r5 from [1, $| C_i^t |$], where $| C_i^t |$ represents the size of candidate nodes. By the way, why the author chooses 5 candidates to mutate? There is no stated evidence to determine five.
2. In Section 4.2.2., the author says that DE algorithm guaranteed convergence, however, metaheuristics such as DE do not guarantee an optimal solution is ever found. Some sufficient conditions must be verified before claiming (global) convergence.
3. In Section 4.2.2., the evaluation score is suspected to be wrong. Because when true y = 1, the successful prediction with \hat{y}=1 yields score as 0, while the failed prediction with \hat{y}=0 has a score as 1. It is strange to see when true y = 0, the positive result (\hat{y}=0) is scored as 1, wheras the score of negative prediction is equal to 2. The scores are inconsistent and biased, without much explanation.
4. In the evaluation step, the author states to implement SVM on every mutation. Considering there are $p$ condidates and $et$ iterations, it will be very time-consuming and not scalable to large number of graphs.
5. In the abstract, it was claimed that the paper aims to address the low interpretability of GNN. However, in the main context, it is not clear which characteristic of the proposed method can address the problem. And there is also no numerical result to support the claim.
6. It seems that the baselines OCGIN and OCGTL are unsupervised methods. It is not clear how the authors perform them in comparison to the proposed semi-supervised method.
7. It is not clear why the authors didn't use AUC as an evaluation metric. The F1 score, precision, and recall are not good enough for imbalanced classification and usually depend on a classification threshold.

**Summary Of The Paper:**

The paper proposed a method for graph-level anomaly detection using the idea of evolutionary mapping. The method is a semi-supervised learning method. The numerical results verified the effectiveness of the proposed method.

**Summary Of The Review:**

Some important points of the paper were not clearly explained. Currently, it is a borderline paper.

---

> ### Author Response · Authors · 2022-11-13
> **Author Response to Reviewer BQDF (1/3)**
>
> Dear Reviewer BQDF,
>
> We sincerely appreciate your insightful and constructive comments. We hope our response can clearly address your concerns and questions. For clarity, we have divided your comments into several questions and our response is as follows:
>
> **Q1:** There are some confusing notations in Section 4.2.2. For informative node selection, the author states the top $k$ nodes most similar to normal graph, but only gives the similarity calculation between a node $j$ and a graph $G_i$, with no clue how node score is aggregated (by summation?).
>
> **R1:** Thanks very much for your reminder. Yes, the overall score of each node is eventually measured as the sum of its similarity score with each normal graph. Accordingly, we revised the manuscript with additional explanations in subsection informative node selection in Section 4.2.2.
>
> *For each node $j$, we take the sum of its similarity scores with all normal graphs as its final score. Eventually, the top $k$ nodes with the highest scores become the initial candidate $\mathbb{C}_{init}$ leveraged in the following DE algorithm.*
>
> **Q2:** For candidate initialization, rigorously speaking, it should be $\\{0, 1\\}^{|\mathbb{C}_i|}$ not $\\{0, 1\\}^{|\mathbb{C}_p|}$.
>
> **R2:** Actually, each candidate selection matrix $I^0_i$ aims to select a subset of nodes from the top $k$ most informative nodes, which means each $diag(I_i^0) \in \mathbb{I}^{0}$ has the same length $|\mathbb{C}_{init}|$. Regarding your comment and to clear up misunderstandings, we have modified the notion as $diag(I^0\_i) \in \\{ 0, 1 \\} ^ { |\mathbb{C}\_{init}| } $.
>
> **Q3:** The parameter $p_s$ in Bernoulli distribution is unstated in the experiment setting.
>
> **R3:** Thanks very much for the comment. We have added the value of parameter $p_s=0.5$ in our experimental settings.
>
> **Q4:** For updating a set of candidates, the subscript of diagonal matrix $I^t$ is supposed to range from 1 to $p$, according to the (1) Initialization. However, the author claims to select $r1$, $r2$, $r3$, $r4$, $r5$ from [1, $|\mathbb{C}_i^{t}|$], where $|\mathbb{C}_i^{t}|$ represents the size of candidate nodes.
>
> **R4:** We sincerely appreciate your correction. $r1$, $r2$, $r3$, $r4$ and $r5$ should be selected from [1, $p$] and we have fixed this typo in the manuscript.
>
> **Q5:** By the way, why the author chooses 5 candidates to mutate? There is no stated evidence to determine five.
>
> **R5:** We adopt five candidates for mutation because this is a commonly used mutation strategy and it has been validated to be effective in generating candidates while maintaining the diversity of candidates in previous research in DE algorithms [1][2]. Considering your concern, we re-clarified this in the manuscript:
>
> *$r1$, $r2$, $r3$, $r4$, and $r5$ are divergent integers randomly selected between $[1, p]$ to maintain the diversity of candidates, as specified in existing works ([1][2]).*
>
> [1] Ahmad, M. F., et al. Differential evolution: A recent review based on state-of-the-art works. Alexandria Engineering Journal, 2021.
>
> [2] Deng, W., et al. An improved differential evolution algorithm and its application in optimization problem. Soft Computing, 2021.
>
> **Q6:** In Section 4.2.2., the author says that DE algorithm guaranteed convergence, however, metaheuristics such as DE do not guarantee an optimal solution is ever found. Some sufficient conditions must be verified before claiming (global) convergence.
>
> **R6:** It is true that DE algorithm can approximate a global optimal solution. In fact, the convergence of DE algorithm has already been theoretically studied in [3]. In this work, as presented in Section 4.2.2, each element in $diag(I_{new}^{t})$ is randomly drawn from $[0,1]$ following the uniform distribution, and this mutation operator satisfies the convergence condition (uniform mutation) validated in [3].
>
> [3] Hu, Z., et al. Sufficient conditions for global convergence of differential evolution algorithm. Journal of Applied Mathematics, 2013.
>
> **Q7:** In Section 4.2.2., the evaluation score is suspected to be wrong. Because when true $y$ = 1, the successful prediction with $\hat{y}=1$ yields score as 0, while the failed prediction with $\hat{y}=0$ has a score as 1. It is strange to see when true $y$ = 0, the positive result ($\hat{y}=0$) is scored as 1, wheras the score of negative prediction is equal to 2. The scores are inconsistent and biased, without much explanation.
>
> **R7:** We thank you very much for the correction. We have corrected our typos in Eq(4). And the second term should be $max(0, \hat{y}\_{G_k})$, $y\_{G_k} = 0 $.

---

> > ### Comment · Reviewer_BQDF · 2022-12-07
> > **After reading the rebuttal**
> >
> > Thanks for the detailed response. The improvement in terms of AUC is not significant enough. By the way, the semi-supervised learning scheme is not that convincing, especially compared with purely supervised learning. Therefore I keep the rating unchanged.

---

> > > ### Author Response · Authors · 2022-12-07
> > > **Response to reviewer BQDF - rebuttal (1/2)**
> > >
> > > Dear Reviewer BQDF,
> > >
> > > Thanks very much for your valuable time and response. We hope our response can address your concerns.
> > >
> > > **Q1:** The improvement in terms of AUC is not significant enough.
> > >
> > > **R1:** From the additional experimental results regarding AUC, we can see that our method performs the best on 7 (out of 9) real-world datasets and the second best on REDDIT-BINARY. And as reported in our paper, our proposed method achieves significantly better Precision, Recall and F1 scores compared to the baselines on all datasets. Hence, taking an overall consideration of those four mostly used evaluation metrics for graph anomaly detection [1,2], our method is better than the 8 latest baselines on graph-level anomaly detection.
> > >
> > > **Q2:** By the way, the semi-supervised learning scheme is not that convincing, especially compared with purely supervised learning.
> > >
> > > **R2:** We sincerely appreciate your comment on the semi-supervised learning scheme. In fact, the lack of labeled anomalies is one of the most significant challenges in graph anomaly detection because identifying all ground-truth anomalies relies heavily on domain expert knowledge and acquiring such data is extremely time-consuming and labor-intensive [1-3]. In most real cases, we may only have the label information of a very limited number of anomalies (a subset of all anomalies) and the detection method should be capable of learning the normal and anomalous distributions from such data effectively. Giving this, most graph anomaly detection methods are developed based on semi-supervised learning scheme [1,2]. Our work follows this scheme to tackle the detection problem in real scenarios and attempts to fully utilize the limited number of labeled data for better anomalous graph detection.
> > >
> > > [1] Akoglu, Leman, Hanghang Tong, and Danai Koutra. Graph based anomaly detection and description: a survey. Data mining and knowledge discovery, 2015.
> > >
> > > [2] Ma, Xiaoxiao, et al. A comprehensive survey on graph anomaly detection with deep learning. IEEE Transactions on Knowledge and Data Engineering, 2021.
> > >
> > > [3] Noble, Caleb C., and Diane J. Cook. Graph-based anomaly detection. Proceedings of the ninth ACM SIGKDD international conference on Knowledge discovery and data mining, 2003.
> > >
> > > We sincerely appreciate your response.
> > >
> > > Best Regards,
> > >
> > > Authors

---

> > > ### Author Response · Authors · 2022-12-08
> > > **Response to reviewer BQDF - rebuttal (2/2)**
> > >
> > > Dear Reviewer BQDF,
> > >
> > > Regarding your concern about the significance of improvement in terms of AUC, we conduct further two-tailed $t$-test (with a $95$ percent level of confidence, $\alpha = 0.05$) between our method (GmapAD-GCN is denoted as A and GmapAD-GAT is denoted as B) and all baselines to show the statistical significance. The pairwise $t$-test $p$-values are reported in the following tables, specifically, each value in the tables is the $p$-value of the pairwise $t$-test between our developed model and the baselines (e.g., A-GCN means the $t$-test between GmapAD-GCN and GCN). According to statistical theory, our method has achieved a statistically significant improvement compared to baselines if the $p$-value is less than $0.05$. And from the tables, we can see that the improvement is statistically significant in terms of AUC since most of the $p$-values are less than $0.05$. We hope this can address your concern.
> > >
> > > |**$p$-value**|**A-GCN**|**A-GAT**|**A-g-U-Nets**|**A-DiffPool**|**A-SAGPool**|**A-GMT**|**A-OCGIN**|**A-OCGLT**|
> > > |-----------------|:----------------:|:----------------:|:---------------------:|:---------------------:|:--------------------:|:-----------------:|:------------------:|:------------------:|
> > > |KKI|4.7e-2|3.1e-2|6.6e-4|4.0e-2|4.3e-2|1.5e-2|4.2e-2|3.0e-3|
> > > |OHSU|2.9e-2|6.7e-4|1.6e-2|4.7e-2|4.1e-2|1.8e-2|2.2e-2|3.6e-6|
> > > |AIDS|3.3e-5|6.0e-4|4.8e-7|3.3e-9|9.5e-1|2.1e-4|1.1e-7|5.1e-7|
> > > |MUTAG|7.7e-7|5.3e-6|1.3e-2|1.6e-2|1.7e-4|5.9e-5|8.4e-4|8.3e-13|
> > > |Mutagenicity|7.6e-16|1.0e-3|4.2e-3|8.2e-4|3.1e-3|5.5e-3|2.1e-6|1.8e-5|
> > > |NCI1|8.7e-4|2.6e-2|6.6e-2|8.7e-2|9.6e-2|3.9e-2|2.4e-7|2.1e-9|
> > > |PROTEINS|1.5e-4|1.4e-8|1.1e-2|3.4e-6|7.6e-3|4.4e-7|2.1e-9|7.5e-11|
> > > |IMDB-BINARY|1.4e-8|3.3e-5|4.1e-3|6.3e-4|2.5e-3|1.2e-3|1.5e-2|2.7e-13|
> > > |REDDIT-BINARY|5.4e-5|1.2e-7|2.8e-2|6.6e-3|5.6e-2|1.9e-1|8.3e-2|1.7e-6|
> > >
> > > |**$p$-value**|**B-GCN**|**B-GAT**|**B-g-U-Nets**|**B-DiffPool**|**B-SAGPool**|**B-GMT**|**B-OCGIN**|**B-OCGLT**|
> > > |-----------------|:----------------:|:----------------:|:---------------------:|:---------------------:|:--------------------:|:-----------------:|:------------------:|:------------------:|
> > > |KKI|6.8e-1|2.5e-2|1.4e-2|1.0e-2|3.6e-1|6.7e-1|1.2e-2|6.0e-3|
> > > |OHSU|1.9e-1|0.8e-4|3.6e-1|9.8e-1|2.4e-1|3.2e-1|2.6e-4|9.3e-5|
> > > |AIDS|2.6e-2|8.1e-1|2.7e-4|2.4e-4|8.2e-4|2.1e-4|3.4e-4|2.4e-4|
> > > |MUTAG|1.6e-3|1.3e-4|1.7e-8|2.4e-8|4.9e-8|3.7e-6|1.6e-2|1.3e-14|
> > > |Mutagenicity|1.3e-5|4.6e-2|3.4e-1|1.1e-1|6.1e-1|2.2e-2|4.8e-6|3.1e-5|
> > > |NCI1|6.7e-4|6.8e-5|5.4e-3|6.6e-3|7.2e-3|6.1e-4|3.4e-9|1.9e-12|
> > > |PROTEINS|3.1e-4|1.1e-14|2.4e-9|8.1e-4|1.9e-7|3.4e-9|7.2e-9|7.2e-13|
> > > |IMDB-BINARY|1.9e-5|8.5e-4|1.1e-3|1.1e-2|2.4e-2|8.9e-3|4.3e-2|4.4e-16|
> > > |REDDIT-BINARY|2.8e-3|7.4e-13|3.7e-6|7.1e-3|1.9e-1|7.3e-3|4.1e-2|3.6e-6|
> > >
> > > Best Regards,
> > >
> > > Authors

---

> ### Author Response · Authors · 2022-11-13
> **Author Response to Reviewer BQDF (2/3)**
>
> **Q8:** In the evaluation step, the author states to implement SVM on every mutation. Considering there are $p$ condidates and $et$ iterations, it will be very time-consuming and not scalable to large number of graphs.
>
> **R8:** Thanks very much for your constructive comment. In this work, we for the first time introduced DE algorithm for graph anomaly detection and we adopt SVM in the evaluation step to ensure that anomalies and normal graphs are well separated in the learned feature space. In fact, integrating SVM with DE is a popular solution for solving optimization problems [4,5,6,7]. And since our topic is graph anomaly detection, we emphasize less optimizing the cost introduced by DE in this paper, but we can design other cost-effective functions to evaluate the fitness of candidates, and our proposed framework is general and can be easily extended to work jointly with them. As motivated by your insightful comments, reducing time costs and handling large graph sets will be an important future research direction for us.
>
> [4] Anton, N., et al. Assessing changes in diabetic retinopathy caused by diabetes mellitus and glaucoma using support vector machines in combination with differential evolution algorithm. Applied Sciences, 2021.
>
> [5] Wang, F. K., et al. One‐sided control chart based on support vector machines with differential evolution algorithm. Quality and Reliability Engineering International, 2019.
>
> [6] Baig, M. Z., et al. Differential evolution algorithm as a tool for optimal feature subset selection in motor imagery EEG. Expert Systems with Applications, 2017.
>
> [7] Lu, X., et al. Classification-assisted differential evolution for computationally expensive problems. In IEEE congress of evolutionary computation, 2011.
>
> **Q9:** In the abstract, it was claimed that the paper aims to address the low interpretability of GNN. However, in the main context, it is not clear which characteristic of the proposed method can address the problem. And there is also no numerical result to support the claim.
>
> **R9:** We sincerely appreciate your comment on the interpretability of the learned graph representations. Our proposed graph mapping module intuitively represents each graph by directly measuring its similarities with a set of candidate nodes that are carefully selected from the graph set, and this has been presented in Section 4.2.1. Regarding your concerns and to clearly state the interpretability of our graph mapping method, we have retitled Section 4.2.1 as Explainable Graph Mapping and added more explanations.
>
> - Section 4.2.1 *Compared to conventional GNN-based models that generate graph-level representations through black-box neural networks, this intuitive graph mapping process is traceable and explainable. The value of each dimension in the graph representation explicitly denotes the proximity between the graph and a specific candidate node. Each candidate node, whilst, is ultimately selected from the graph set and can be traced back to its belonging graph, which is achieved by our specially designed DE algorithm detailed in the following section.*
>
> - Section 4.2.2 *This intuitive binary selection process directly chooses nodes from the graph set and the selected node can be easily traced back to its belonging graph.*
>
> **Q10:** It seems that the baselines OCGIN and OCGTL are unsupervised methods. It is not clear how the authors perform them in comparison to the proposed semi-supervised method.
>
> **R10:** In fact, OCGIN and OCGTL are one-class methods. Both of them only take normal graphs for learning a hypersphere to identify anomalies and normal graphs, by which they use label information for training.

---

> ### Author Response · Authors · 2022-11-13
> **Author Response to Reviewer BQDF (3/3)**
>
> **Q11:** It is not clear why the authors didn't use AUC as an evaluation metric. The F1 score, precision, and recall are not good enough for imbalanced classification and usually depend on a classification threshold.
>
> **R11:** Thanks very much for your comment. Per your suggestion, we conduct additional tests and report the results on AUC as follows. We can see that GmapAD in most cases performs better than the baselines.
>
> |**AUC**|**GCN**|**GAT**|**g-U-Nets**|**DiffPool**|**SAGPool**|**GMT**|**OCGIN**|**OCGLT**|**GmapAD-GCN**|**GmapAD-GAT**|
> |-----------------|:----------------:|:----------------:|:---------------------:|:---------------------:|:--------------------:|:-----------------:|:------------------:|:------------------:|:-----------------------:|:-----------------------:|
> |KKI|0.52 $\pm$ 0.1|0.44 $\pm$ 0.2|0.35 $\pm$ 0.2|0.50 $\pm$ 0.2|0.50 $\pm$ 0.1|0.52 $\pm$ 0.1|0.44 $\pm$ 0.1|0.35 $\pm$ 0.1|**0.64 $\pm$ 0.2**|0.55 $\pm$ 0.2|
> |OHSU|0.62 $\pm$ 0.2|0.51 $\pm$ 0.1|0.53 $\pm$ 0.1|0.59 $\pm$ 0.2|0.53 $\pm$ 0.2|0.57 $\pm$ 0.2|0.50 $\pm$ 0.2|0.46 $\pm$ 0.2|**0.69 $\pm$ 0.2**|0.64 $\pm$ 0.3|
> |AIDS|0.64 $\pm$ 0.1|0.66 $\pm$ 0.1|**0.98 $\pm$ 0.1**|0.98 $\pm$ 0.1|0.93 $\pm$ 0.1|**0.98 $\pm$ 0.1**|0.96 $\pm$ 0.1|0.97 $\pm$ 0.1|0.93 $\pm$ 0.1|0.92 $\pm$ 0.1|
> |MUTAG|0.54 $\pm$ 0.1|0.50 $\pm$ 0.1|0.76 $\pm$ 0.1|0.76 $\pm$ 0.1|0.73 $\pm$ 0.1|0.72 $\pm$ 0.1|0.63 $\pm$ 0.4|0.66 $\pm$ 0.1|**0.79 $\pm$ 0.1**|0.73 $\pm$ 0.2|
> |Mutagenicity|0.46 $\pm$ 0.1|0.65 $\pm$ 0.1|0.68 $\pm$ 0.1|0.62 $\pm$ 0.1|0.68 $\pm$ 0.1|**0.69 $\pm$ 0.1**|0.47 $\pm$ 0.1|0.55 $\pm$ 0.1|**0.69 $\pm$ 0.1**|0.67 $\pm$ 0.1|
> |NCI1|0.62 $\pm$ 0.1|0.62 $\pm$ 0.1|0.65 $\pm$ 0.1|0.66 $\pm$ 0.1|**0.67 $\pm$ 0.1**|0.66 $\pm$ 0.1|0.35 $\pm$ 0.1|0.63 $\pm$ 0.1|**0.67 $\pm$ 0.1**|**0.67 $\pm$ 0.1**|
> |PROTEINS|0.65 $\pm$ 0.1|0.61 $\pm$ 0.1|0.83 $\pm$ 0.1|0.73 $\pm$ 0.1|**0.88 $\pm$ 0.1**|0.66 $\pm$ 0.1|0.39 $\pm$ 0.1|0.69 $\pm$ 0.1|**0.88 $\pm$ 0.1**|0.79 $\pm$ 0.2|
> |IMDB-BINARY|0.67 $\pm$ 0.1|0.69 $\pm$ 0.1|0.76 $\pm$ 0.1|0.69 $\pm$ 0.1|0.85 $\pm$ 0.1|0.77 $\pm$ 0.1|0.61 $\pm$ 0.3|0.67 $\pm$ 0.3|**0.87 $\pm$ 0.1**|0.81 $\pm$ 0.1|
> |REDDIT-BINARY|0.73 $\pm$ 0.1|0.68 $\pm$ 0.1|0.74 $\pm$ 0.1|0.74 $\pm$ 0.1|0.75 $\pm$ 0.1|**0.78 $\pm$ 0.1**|0.42 $\pm$ 0.4|0.75 $\pm$ 0.2|0.76 $\pm$ 0.1|0.73 $\pm$ 0.1|

---

### Official Review · Reviewer_H8Au · 2022-10-25

**Confidence:** 3
**Correctness:** 4
**Technical Novelty And Significance:** 3
**Empirical Novelty And Significance:** 3
**Recommendation:** 6

**Clarity, Quality, Novelty And Reproducibility:**

Regarding reproducibility, the authors claimed that they provided the full source code for their experiments, however, the link they provided pointed to an empty project.


**Strength And Weaknesses:**

This appears to be a well-researched paper.
The authors clearly defined the problem, motivated their solution and tested their approach
on nine benchmark datasets. They also performed ablation tests that really help illustrating the importance of each major step in their pipeline.
In the appending the authors discussed their parameter choices their made for DE algorithm,

**Summary Of The Paper:**

In this papers the authors proposed a new methodology for finding anomalies in graph data. The focus of this paper is on detecting anomalies using graph-level classification.
Graph-level anomaly detection is not a new problem. The authors claimed that previous literature has not addressed both intra-graph and  inter-graph relationships as being important features for finding anomalous graphs.
This paper clearly explained the issues with the previous methods and pointed to two important aspects lacking in previous solutions. First aspect had to do with not taking into consideration inter-graph relations when finding anomalous graphs. Previous methods focused primarily on intra-graph relations. The second aspect is the GNN's inherent lack of explainability.
To address both problems the authors introduced a pipeline that would link graphs (neighborhood nodes aggregated representations) with the most informative set of nodes (taken from all graphs). They would then transform the graphs into candidate feature space and find the best candidate set by applying differential evolution algorithm. The end stage of their pipeline involves running a classifier on graphs in candidate feature space.
As a result the authors claim to improve both explainability and model efficacy.

**Summary Of The Review:**

Overall, I found this paper to be well-written and well-motivated. I believe it addressed an important problem and provided an interesting solution.

I would be interested to learn more about measuring the explaiability of approaches. We know that some methods are black boxes and others can be easily interpreted, but this leaves a lot of ground in between. It would be useful if interpretability could also be expressed more precisely.

Also, I would like to ask the authors about their for the future work. Is there there something in their current method they would like to improve upon?

On more question. Could the authors explain how the negative effect of the curse of dimensionality gets avoided during the informative node selection stage?

---

> ### Author Response · Authors · 2022-11-13
> **Author Response to Reviewer H8Au**
>
> Dear Reviewer H8Au,
>
> We greatly thank you for your valuable time and support of the importance of our research, our motivation, paper presentation and the proposed framework GmapAD. We hope our response can address your questions.
>
> **Q1:** Regarding reproducibility, the authors claimed that they provided the full source code for their experiments, however, the link they provided pointed to an empty project.
>
> **R1:** Thank you very much for pointing out the out-of-date anonymous URL. We have re-linked our code repository with the provided URL and our source code is now accessible.
>
> **Q2:** Overall, I found this paper to be well-written and well-motivated. I believe it addressed an important problem and provided an interesting solution. I would be interested to learn more about measuring the explaiability of approaches. We know that some methods are black boxes and others can be easily interpreted, but this leaves a lot of ground in between. It would be useful if interpretability could also be expressed more precisely.
>
> **R2:** We greatly thank you for your confirmation and feedback. To clearly state the interpretability of our graph mapping method, we have retitled Section 4.2.1 as Explainable Graph Mapping and added more explanations.
>
> - Section 4.2.1 *Compared to conventional GNN-based models that generate graph-level representations through black-box neural networks, this intuitive graph mapping process is traceable and explainable. The value of each dimension in the graph representation explicitly denotes the proximity between the graph and a specific candidate node. Each candidate node, whilst, is ultimately selected from the graph set and can be traced back to its belonging graph, which is achieved by our specially designed DE algorithm detailed in the following section.*
>
> - Section 4.2.2 *This intuitive binary selection process directly chooses nodes from the graph set and the selected node can be easily traced back to its belonging graph.*
>
> **Q3:** Also, I would like to ask the authors about their future work. Is there there something in their current method they would like to improve upon?
>
> **R3:** We sincerely appreciate your interests in our future work. As can be seen, in this work, we for the first time introduced DE algorithm for learning graph-level representations and because our research interest is on graph anomaly detection, we only adopted the very basic but validated DE algorithm for selecting and optimizing candidate nodes. There are extensive researches in optimizing DE algorithms for handling large-scale data and reducing the time cost. In the future, we may adopt more advanced DE algorithms and propose better fitness scoring functions for DE to explore more effective solutions. And to be highlighted, the framework proposed in this work is flexible to be integrated to other DE algorithms and GNNs to empower its industrial applications.
>
> **Q4:** One more question. Could the authors explain how the negative effect of the curse of dimensionality gets avoided during the informative node selection stage?
>
> **R4:** It is our pleasure to explain this. Directly mapping each graph regarding its similarity with all nodes in the graph set results in extremely high dimensional graph-level representations because of the huge number of nodes in the set. Regarding the curse of dimensionality, if we apply such a straightforward method, each graph in the representation space will be different from others and anomalies can not be effectively identified. Our informative node selection stage attempts to address this by reducing number of nodes utilized for learning graph representations, which is achieved by selecting the top $k$ nodes that are most similar to all normal graphs.
>
> Best Regards,
>
> Authors

---

### Official Review · Reviewer_eKPL · 2022-11-04

**Confidence:** 3
**Correctness:** 3
**Technical Novelty And Significance:** 2
**Empirical Novelty And Significance:** 2
**Recommendation:** 5

**Clarity, Quality, Novelty And Reproducibility:**

Clarity: Good

Reproducibility: Excellent

Quality: Good

Novelty: Medium

**Strength And Weaknesses:**

Strengths

- The idea of node candidate optimization via differential evolutionary algorithm is interesting.

- Experimental results show superior performances of GmapAD compared to existing GNN-based graph-level anomaly detection models.

Weaknesses

- In Eq. (1), the author simply uses the average pooling as a graph pooling operation. But, If using attention-based pooling operations ([1], [2], [3]), I think the globally informative node selection is possible with GNN itself without additional candidate optimization. It would be good to have a comparison between GNNs with attention-based pooling and GNNs with candidate optimization.

- I cannot agree with the author’s statement “Most importantly, simply reading out graph representations using its own nodes cannot capture the inter-graph information.”.  Even though each graph has different nodes, since the features of nodes are in the same space, inter-graph information can be captured with graph representations from GNNs with a readout function.

- The author stated that interpretability is an advantage of a framework that proposes interpretability, but there is no analysis of this.

[1] Gao, Hongyang, and Shuiwang Ji. "Graph u-nets." international conference on machine learning. PMLR, 2019.

[2] Lee, Junhyun, Inyeop Lee, and Jaewoo Kang. "Self-attention graph pooling." International conference on machine learning. PMLR, 2019.

[3] Ying, Zhitao, et al. "Hierarchical graph representation learning with differentiable pooling." Advances in neural information processing systems 31 (2018).


**Summary Of The Paper:**

This paper presents a GNN-based framework for graph-level anomaly detection, GmapAD. Specifically, GmapAD encapsulate graph structures and node features into node representations and selects the best subset of informative nodes in all possible nodes in graphs for mapping graphs into the new feature space. Then normal and anormal graphs are mapped to deviating locations in the feature space and classified using machine learning techniques (e.g., SVM). Experiments conducted on 9 benchmarks show the effectiveness of the proposed framework compared to state-of-the-art GNN models and graph-level anomaly detection methods.


**Summary Of The Review:**

This paper presents a graph-level anomaly detection framework. The presentation of the paper is clear and easy to follow, and the experimental result is convincing, but the motivation is unclear.

---

> ### Author Response · Authors · 2022-11-13
> **Author Response to Reviewer eKPL (1/2)**
>
> Dear Reviewer eKPL,
>
> We thank you very much for your confirmation of the node optimization idea, clarity, reproducibility, quality and experimental results of this work and valuable and constructive reviews. We have addressed all your detailed concerns with additional experiments on three listed baselines and we sincerely hope that our response can present the motivation of this work more clearly. Our responses are as follows:
>
> **Q1:** In Eq. (1), the author simply uses the average pooling as a graph pooling operation. But, If using attention-based pooling operations ([1], [2], [3]), I think the globally informative node selection is possible with GNN itself without additional candidate optimization. It would be good to have a comparison between GNNs with attention-based pooling and GNNs with candidate optimization.
>
> **R1:** We agree that graph pooling is vital in graph-level anomaly detection and comparison with graph pooling methods could better demonstrate the effectiveness of our framework. Following your suggestion, we have conducted additional experiments by comparison with the three listed baselines, i.e., g-U-nets [1], SAGPool [2] and DIFFPOOL [3]. We can see that GmapAD constantly performs better than the three added baselines. We have also revised the manuscript with updated test results and other related sections.
>
> |**Precision**|**g-U-Nets[1]**|**DiffPool[3]**|**SAGPool[2]**|**GmapAD-GCN**|**GmapAD-GAT**|
> |---------------|:-----------------------:|:-----------------------:|:----------------------:|:-------------------------:|:-------------------------:|
> |KKI|0.38 $\pm$ 0.2|0.46 $\pm$ 0.1|0.36 $\pm$ 0.2|0.63 $\pm$ 0.3|**0.67 $\pm$ 0.4**|
> |OHSU|0.51 $\pm$ 0.3|0.48 $\pm$ 0.2|0.46 $\pm$ 0.2|0.59 $\pm$ 0.1|**0.66 $\pm$ 0.2**|
> |AIDS|0.97 $\pm$ 0.1|0.97 $\pm$ 0.1|0.92 $\pm$ 0.1|**0.98 $\pm$ 0.1**|**0.98 $\pm$ 0.1**|
> |MUTAG|0.88 $\pm$ 0.1|0.75 $\pm$ 0.1|0.80 $\pm$ 0.1|**0.93 $\pm$ 0.3**|0.84 $\pm$ 0.2|
> |Mutagenicity|0.88 $\pm$ 0.1|0.72 $\pm$ 0.1|0.88 $\pm$ 0.1|0.90 $\pm$ 0.1|**0.91 $\pm$ 0.1**|
> |NCI1|0.89 $\pm$ 0.1|0.61 $\pm$ 0.1|0.91 $\pm$ 0.1|**0.93 $\pm$ 0.2**|**0.93 $\pm$ 0.3**|
> |PROTEINS|0.87 $\pm$ 0.3|0.75 $\pm$ 0.1|0.87 $\pm$ 0.1|**0.89 $\pm$ 0.1**|0.87 $\pm$ 0.2|
> |IMDB-BINARY|0.94 $\pm$ 0.1|0.76 $\pm$ 0.1|0.90 $\pm$ 0.1|**0.98 $\pm$ 0.1**|0.92 $\pm$ 0.1|
> |REDDIT-BINARY|0.89 $\pm$ 0.1|0.74 $\pm$ 0.1|0.91 $\pm$ 0.1|**0.93 $\pm$ 0.1**|0.91 $\pm$ 0.1|
>
> |**Recall**|**g-U-Nets[1]**|**DiffPool[3]**|**SAGPool[2]**|**GmapAD-GCN**|**GmapAD-GAT**|
> |-----------------|:---------------------:|:---------------------:|:--------------------:|:-----------------------:|:-----------------------:|
> |KKI|0.32 $\pm$ 0.1|0.46 $\pm$ 0.1|0.68 $\pm$ 0.5|**0.83 $\pm$ 0.3**|0.80 $\pm$ 0.4|
> |OHSU|0.45 $\pm$ 0.4|0.49 $\pm$ 0.2|0.65 $\pm$ 0.2|**0.82 $\pm$ 0.2**|0.64 $\pm$ 0.2|
> |AIDS|0.73 $\pm$ 0.2|0.93 $\pm$ 0.1|0.82 $\pm$ 0.1|**0.99 $\pm$ 0.1**|**0.99 $\pm$ 0.1**|
> |MUTAG|0.85 $\pm$ 0.1|0.80 $\pm$ 0.1|0.82 $\pm$ 0.1|**0.95 $\pm$ 0.3**|0.89 $\pm$ 0.2|
> |Mutagenicity|0.95 $\pm$ 0.2|0.74 $\pm$ 0.1|0.95 $\pm$ 0.1|**0.98 $\pm$ 0.1**|**0.98 $\pm$ 0.1**|
> |NCI1|0.96 $\pm$ 0.1|0.58 $\pm$ 0.1|0.95 $\pm$ 0.1|**0.99 $\pm$ 0.1**|**0.99 $\pm$ 0.1**|
> |PROTEINS|0.71 $\pm$ 0.1|0.68 $\pm$ 0.1|0.92 $\pm$ 0.1|**0.99 $\pm$ 0.1**|0.97 $\pm$ 0.2|
> |IMDB-BINARY|0.98 $\pm$ 0.1|0.77 $\pm$ 0.1|0.98 $\pm$ 0.1|**0.99 $\pm$ 0.1**|0.94 $\pm$ 0.1|
> |REDDIT-BINARY|0.78 $\pm$ 0.2|0.68 $\pm$ 0.1|0.91 $\pm$ 0.1|0.94 $\pm$ 0.1|**0.96 $\pm$ 0.1**|
>
> |**F1**|**g-U-Nets[1]**|**DiffPool[3]**|**SAGPool[2]**|**GmapAD-GCN**|**GmapAD-GAT**|
> |-----------------|:---------------------:|:---------------------:|:--------------------:|:---------------------:|:-----------------------:|
> |KKI|0.34 $\pm$ 0.1|0.44 $\pm$ 0.1|0.46 $\pm$ 0.3|0.71 $\pm$ 0.3|**0.73 $\pm$ 0.3**|
> |OHSU|0.36 $\pm$ 0.3|0.47 $\pm$ 0.2|0.51 $\pm$ 0.2|**0.68 $\pm$ 0.7**|0.63 $\pm$ 0.2|
> |AIDS|0.83 $\pm$ 0.1|0.98 $\pm$ 0.1|0.86 $\pm$ 0.1|**0.99 $\pm$ 0.1**|**0.99 $\pm$ 0.1**|
> |MUTAG|0.82 $\pm$ 0.1|0.83 $\pm$ 0.1|0.81 $\pm$ 0.1|**0.96 $\pm$ 0.3**|0.86 $\pm$ 0.1|
> |Mutagenicity|0.92 $\pm$ 0.1|0.72 $\pm$ 0.1|0.91 $\pm$ 0.1|**0.94 $\pm$ 0.1**|**0.94 $\pm$ 0.1**|
> |NCI1|0.94 $\pm$ 0.1|0.59 $\pm$ 0.1|0.95 $\pm$ 0.1|0.96 $\pm$ 0.2|**0.97 $\pm$ 0.2**|
> |PROTEINS|0.78 $\pm$ 0.1|0.71 $\pm$ 0.1|0.92 $\pm$ 0.1|**0.94 $\pm$ 0.1**|0.93 $\pm$ 0.2|
> |IMDB-BINARY|0.94 $\pm$ 0.1|0.91 $\pm$ 0.2|0.94 $\pm$ 0.1|**0.95 $\pm$ 0.1**|0.94 $\pm$ 0.1|
> |REDDIT-BINARY|0.82 $\pm$ 0.1|0.69 $\pm$ 0.1|0.93 $\pm$ 0.1|0.94 $\pm$ 0.1|**0.95 $\pm$ 0.1**|
>
> Reference:
>
> [1] Gao, Hongyang, and Shuiwang Ji. Graph u-nets. international conference on machine learning. PMLR, 2019.
>
> [2] Lee, Junhyun, Inyeop Lee, and Jaewoo Kang. Self-attention graph pooling. International conference on machine learning. PMLR, 2019.
>
> [3] Ying, Zhitao, et al. Hierarchical graph representation learning with differentiable pooling. Advances in neural information processing systems, 2018.

---

> ### Author Response · Authors · 2022-11-13
> **Author Response to Reviewer eKPL (2/2)**
>
> **Q2:** I cannot agree with the author’s statement “Most importantly, simply reading out graph representations using its own nodes cannot capture the inter-graph information.”. Even though each graph has different nodes, since the features of nodes are in the same space, inter-graph information can be captured with graph representations from GNNs with a readout function.
>
> **R2:** Thanks for your insightful comment. We agree that inter-graph information could be implicitly captured because the GNN is trained using all graphs. However, such information is encapsulated only in node representations and simply read out graph-level representations using a graph's own nodes (as in most existing works) will inherently ignore useful node information in other graphs.
>
> For graph-level anomaly detection, we seek to learn divergent representations for anomalies and normal graphs. If we can find a benchmark (a set of nodes) from the whole graph set such that anomalies are dissimilar to the benchmark while normal graphs are similar, anomalies can then be effectively detected. To this end, we design GmapAD to explore inter-graph information more comprehensively and directly learn graph representations with regard to a carefully selected benchmark. To clear up misunderstandings, we have revised this statement in the manuscript and we hope our motivation is better and clearly stated through the revision.
>
> *Most importantly, simply reading out graph representations using its own nodes cannot explicitly and effectively capture the inter-graph information.*
>
> **Q3:** The author stated that interpretability is an advantage of a framework that proposes interpretability, but there is no analysis of this.
>
> **R3:** We thank the reviewer's valuable comment on the interpretability of our proposed framework. In fact, we have presented why our proposed graph mapping method is traceable and explainable in Section 4.2.1. Considering the reviewer's concern, we have modified the title of Section 4.2.1 as Explainable Graph Mapping and further explained this by:
>
> - Section 4.2.1 *Compared to conventional GNN-based models that generate graph-level representations through black-box neural networks, this intuitive graph mapping process is traceable and explainable. The value of each dimension in the graph representation explicitly denotes the proximity between the graph and a specific candidate node. Each candidate node, whilst, is ultimately selected from the graph set and can be traced back to its belonging graph, which is achieved by our specially designed DE algorithm detailed in the following section.*
>
> - Section 4.2.2 *This intuitive binary selection process directly chooses nodes from the graph set and the selected node can be easily traced back to its belonging graph.*
>
> **Q4:** This paper presents a graph-level anomaly detection framework. The presentation of the paper is clear and easy to follow, and the experimental result is convincing, but the motivation is unclear.
>
> **R4:** We are very grateful for your encouraging comments on the presentation and experimental result. The motivations behind this work are: 1) We aim to achieve better graph-level anomaly detection through a more comprehensive exploration of *inter-* and *intra-* graph information in the graph set. 2) We seek to learn discriminative representations for anomalies and normal graphs such that anomalies can be more effectively identified. 3) We for the first time adopt a tractable and explainable graph mapping strategy to learn graph representations that can intuitively reflect the correlation between graphs and nodes in the graph set.

---

### Decision · Program_Chairs · 2023-01-20

**Decision:**

Reject

**Justification For Why Not Higher Score:**

As I have stated in the meta-review and the AC-reviewer meeting report, this paper has several crucial concerns, which may change the conclusion of the paper after fixing them. Therefore the current version is not ready for publication. Since the proposed approach is potentially interesting and novel, I strongly recommend revising the paper according to reviewers' comments and re-submitting it.


**Justification For Why Not Lower Score:**

N/A

**Metareview: Summary, Strengths And Weaknesses:**

This paper proposes a graph-level anomaly detection method, called GmapAD. The proposed method performs the standard message passing procedure by considering anomalous labels to incorporate intra-graph information, followed by applying the differential evolutionary mapping to the obtained node representations to obtain graph representations incorporating inter-graph information. Finally, any ML algorithms such as SVM can be applied to the obtained graph representations to detect anomalies. The proposal is empirically evaluated on real-world datasets.

### Strength

- The idea of using differential evolutionary mapping to optimize node candidates to explicitly incorporate inter-graph information is novel and interesting.

- After the authors' rebuttal and revision of the paper, the paper is now clearly written and easy to follow.

### Weakness

- As reviewers pointed out, the interpretability issue should be further elaborated. Currently it is unclear how the proposal is more interpretable than other approaches.

- The problem setting is unclear. The authors mention that the proposal is semi-supervised in the last paragraph in Section 3. However, this setting is never considered afterwards and it seems that any class labels are assumed to be available. Moreover, the standard SVM is used at the final step, which also indicates that the task is fully supervised. This point should be clarified.

- Related to the above point, empirical comparison is unfair. The proposed method and three GNNs use class label information in the node embedding process and the final classification step. However, the two graph-level anomaly detection methods OCGIN and OCGLT work in an unsupervised manner (or partially supervised in the sense that only normal data are available in training), and one-class SVM is used for anomaly detection instead of the standard SVM.

- Important baselines, graph kernels, are missing in experiments. In particular, the Weisfeiler-Lehman graph kernel often shows competitive performance compared to GNN-based methods. Since one can explicitly obtain feature representations of graphs via such graph kernels, it should be considered as one of comparison partners.

**Summary Of Ac-Reviewer Meeting:**

Since this paper is a borderline paper, the AC-reviewer meeting was held, and we have extensively and carefully discussed pros and cons of this paper while considering the authors' response and the revised paper.
Although we agree that this paper is potentially interesting, from reviewers, several concerns were raised, including:

- The interpretability of the proposal is not convincing.
- The proposal can be time-consuming, thus the efficiency should be evaluated.
- Although the authors added results of AUC, which is actually the most standard measure in anomaly detection, the improvement of the proposal is not significant.
- Unclear problem setting (supervised? semi-supervised? how is it considered in experiments?).
- Graph kernels should be compared as baselines.

In particular, as I have written in the meta-review, the interpretability issue and unclear issues in experiments are crucial. We have therefore agreed with rejecting the paper.